# How Efficient Is the Implementation of Structural Funds Committed to Enhancing ICT Adoption in SMEs?

Carla Henriques [1,2,3,*] and Clara Viseu [1,4]

1   Coimbra Business School, Instituto Superior de Contabilidade e Administração de Coimbra (ISCAC), Polytechnic of Coimbra, 3045-601 Coimbra, Portugal
2   INESC Coimbra, Department of Electrical and Computer Engineering (DEEC), University of Coimbra, Polo 2, 3030-290 Coimbra, Portugal
3   Centre for Business and Economics Research (CeBER), Faculty of Economics, University of Coimbra, Av Dias da Silva 165, 3004-512 Coimbra, Portugal
4   Centro de Matemática e Aplicações (CMA-UBI), Universidade da Beira Interior, 6200-001 Covilhã, Portugal
*   Correspondence: chenriques@iscac.pt

**Abstract:** We evaluated the execution of the operational programs (OPs) committed to encouraging the adoption of information and communication technologies (ICTs) in small and medium-sized enterprises (SMEs). To achieve this goal, we employed a novel three-stage weighted Russel directional distance (WRDD) data envelopment analysis (DEA) model in conjunction with stochastic frontier analysis (SFA), which considers indicators officially mandated by the European Union (EU) and environmental factors, to evaluate 51 OPs from 16 EU countries. All in all, we concluded that by removing the environmental factors, about 30% of the OPs (16) reached an efficient procedural performance against 20% (10). The OP more frequently viewed as a benchmark regardless of the environmental factors is "Multi-regional Spain—ERDF" that remains robustly efficient within 5% and 10% tolerances. Without the removal of the environmental factors, the "number of operations supported" is the indicator that requires more attention from management authorities (MAs), whereas with their removal one-third of the OPs need to further reduce the "eligible costs decided" and improve "eligible spending". According to our findings, more developed regions and a higher rate of ICT specialists seem to be related to an underuse of ERDF funds dedicated to boosting ICT in SMEs. These findings might be related to the administrative burden and the lack of ability of SMEs to deal with the various procedures for applying for and implementing European Regional Development Fund (ERDF) projects. Overall, it is critical to provide further support that simplifies administrative procedures and addresses SMEs' specific requirements.

**Keywords:** ICT; EU regions; SMEs; WRDD model; ERDF

**JEL Classification:** C6; C1; O3; M1

## 1. Introduction

In a time of digital transformation, global markets, and pandemics, businesses must increase their internal skills to deal with dynamic environments [1]. Innovative ICTs such as the Internet of Things, mobile devices, big data, and machine learning have a proven impact on organizations' operating contexts, radically altering the business practices of many firms as well as how customers engage with these businesses and other players [2–4]. ICT tools are usually used to gather, save, manage, disseminate, and spread information, bringing together both physical devices (such as computers, networks, and terminals) and the software or applications which allow running them [5].

According to the literature, ICT can assist enterprises in adapting to international markets by lowering border operating costs and allowing wider access to critical innovative resources [6–8]. In addition, it is broadly acknowledged that ICT can provide a

substantial positive impact on firms' performance [3,9,10]. The impact of ICT on improving connectivity both internally and externally is critical to SMEs' successful innovation [11–13]. Moreover, it has been established that using broadband Internet has a favorable influence on SMEs' innovativeness [14]. In addition, different authors have ascertained that ICT may help small enterprises enhance their productivity, efficiency, and performance [15–18]. In addition, several studies revealed that the outset of the COVID-19 turmoil led to a rapid adaptation to the use of digital technologies, especially for SMEs (e.g., [19,20]).

Since SMEs in Europe comprise 99% of all firms, employ around 100 million workers, and are responsible for more than 50% of the EU's gross domestic product [21], ICT adoption in SMEs should be continuously enhanced.

However, despite the advantages and possibilities that ICT and digital technologies may provide, as well as the rapid expansion of their acceptance in the past few years, SMEs are still not fully exploring their potential [22,23]. The transformative potential of emerging technologies, particularly digital technologies and ICT, still poses a challenge to organizations [24–26].

In part, this might be related to the fact that SMEs possess scarce resources, technology, and skills. In effect, there are several barriers to ICT adoption by SMEs [27]:

- Financial: large spending required and difficulties in obtaining credit;
- Infrastructure: electricity costs, broadband, and consistent Internet connectivity;
- Organizational: a shortage of trained personnel;
- Technical: technological growth without proper preparation.

Another factor hampering ICT adoption by SMEs refers to the lack of perception regarding the potentialities and impacts that their digital transformation might bring [28,29]. On the one hand, SMEs risk losing competitiveness, productivity, and viability if they do not undertake digital transformation [30,31]. On the other hand, managers tend to reject digital projects since they do not know how to integrate them into the firm [28,32]. In fact, the question of whether and how innovation can be reproduced and used in a broader framework is inextricably linked to an awareness of the elements and procedures that might determine the (un)successful adoption of the technology itself [33].

To encourage ICT investment in SMEs, it is critical to adopt governmental policies that reduce the digital gap, offer free broadband Internet connectivity, and promote education [28,29,32]. In this context, digital transformation has many definitions [34]. Fitzgerald et al. [35] define digital transformation as the application of new digital technologies such as the Internet, smartphones, analytics, or smart objects to enable large business changes. These in their turn allow for boosting customer satisfaction, optimizing processes, or producing innovative business practices. As a result, digital transformation goes beyond just digitizing resources to generate value and money from digital assets. Similarly, [36] differentiates between digital transformation and digitization. Whereas digitization refers to the simple conversion of analog data into digital data, digital transformation refers to a broader understanding and shift in attitude that influences politics, business, and social concerns. This digital transition, according to the "Plattform Industrie 4.0", depicts a stride forward in which people, equipment, and goods are immediately connected to their surroundings [25].

All things considered, the advancement of ICT is critical for attaining Europe's prosperity, particularly with the continuous development of a digital globalized market. Therefore, investing in ICT projects is fundamental for achieving the European Commission's goal of preparing Europe for the digital era [37].

Over 2000–2006, ICT assistance for SMEs became a major concern, with around EUR 2.3 billion in Cohesion Policy financing, corresponding to 24% of Cohesion Policy grants for digital projects [38]. This contribution reached over EUR 2.6 billion between 2007 and 2013; however, it has since been reduced to EUR 2.1 billion for the period 2014–2020, with just 10% of Cohesion Policy funds committed to digital developments [38].

In the context of Cohesion Policy funds, Member states (MSs) were expected to conduct evaluations of their efficacy, efficiency, and impacts since 2014. Notwithstanding, there

are still concerns that receive insufficient research interest, particularly when evaluations take place during the programmatic time horizon. According to [39], policymakers have to face substantial challenges in designing and implementing research and innovation policies, notably during the monitoring and evaluation stages, mainly due to a lack of appropriate data. The case of EU ICT policies is no exception since researchers have also been highlighting both the lack of scholarly attention and data on the adoption of ICTs in firms at the regional level [5]. Understanding the ICT investment strategies of EU regions is not an easy procedure [40]. Besides being considered an activity sector, ICT is also an integrated element of other linked activity sectors (e.g., e-Health) and a tool to support other activities. Because the actions implemented with European Structural and Investment Funds (ESIF) sometimes have several goals, it is difficult to identify ICT-connected activities within the allotted categories when the OPs are scheduled. The financial data of the OPs are organized into Categories of Intervention, Thematic Objectives (TOs), and priority areas. It is stated in the guideline material for regions and MSs to help them write OPs that planned ICT activities should be categorized predominantly under Thematic Objective 2 (TO2). However, ICT initiatives can also obtain funding within different TOs, and they are also incorporated into several smart specialization policies.

Hence, this study aims to contribute to the literature by proposing a novel overarching methodological framework that allows policymakers (e.g., MAs) to monitor the implementation of OPs devoted to supporting ICT adoption in SMEs through the use of a nonparametric approach. We propose a novel three-stage WRDD model in combination with SFA to evaluate 51 OPs from 16 EU MSs.

The first stage involves employing the WRDD model to calculate each OP's efficiency scores. At this stage, valuable information is computed that provides further understanding of the necessary corrections that need to occur to tackle the potential discrepancies from the OPs viewed as benchmarks. In comparison to other methodologies usually used in this context (e.g., review of benchmark case studies, econometric analysis, statistical analysis, macroeconomic and microeconomic studies), the WRDD model can be notably helpful for MAs, since it allows them to recognize the benchmarks and changes that should be operated to improve the successful implementation of the OPs. The second stage involves applying SFA to the OPs deemed as inefficient to compute the adjusted input and output factors, by removing the significant environmental effects and statistical noises. At this point, it is also possible to identify the main environmental factors that can influence the efficiency of the implementation of ERDF funds in different OPs dedicated to enhancing ICT adoption in SMEs and to see the importance of management failures. Finally, in stage 3, the previously adjusted input and output factors of every inefficient DMU are used to compute new efficiency scores with the WRDD model.

Overall, the key research questions that this work aims to address are as follows:

RQ1. "Which indicators prevent the efficient utilization of ERDF allocated to boost ICT adoption in EU SMEs?"

RQ2: "Which OPs were most frequently referenced as a source of best practices for the programmatic period under analysis?"

RQ3: "Which OPs show more efficiency resilience in the face of probable changes in the indicators used?"

RQ4: "Which environmental factors have the greatest impact on the inefficiency of the execution of OPs aimed at boosting ICT in EU SMEs?"

RQ5: "How does efficiency change with the removal of environmental factors?"

In this perspective, the main novelties of this work are fourfold: (1) it suggests using the WRDD DEA model in conjunction with SFA, proposing a novel three-stage WRDD model approach; (2) it provides a novel approach to conduct for the first time an evaluation strictly targeting the OPs dedicated to fostering ICT in EU SMEs; (3) it helps to identify if the changes required to attain efficiency are linked to management inefficiency or environmental factors, thus offering support that enables shaping the required policies to overcome the detected inefficiencies; (4) it computes efficiency by considering adjusted

factors obtained after removing the impact that environmental factors might have on the OPs' performance.

The structure of this document is as follows: Section 2 includes an overview of the literature on Cohesion Policy evaluations of ICT in SMEs. Section 3 describes the main premises behind the approach proposed to evaluate the implementation of the OPs under scrutiny. Section 4 discusses the major reasons behind the selection of the input and output factors used in the efficiency evaluation, and also some basic statistics on the data that were used to feed the WRDD and SFA models. Section 5 examines the main results. Section 6 presents the key conclusions, offers potential policy implications, highlights the main limitations, and suggests potential future work developments.

## 2. Literature Review

Over the last ten years, several efforts have been undertaken to address the role of ICT adoption in SMEs, leading to the publication of a broad range of review studies. In this context, [41] examined two theoretical models—the diffusion of innovation theory [42] and the technology, organization, and environment structure [43]—to provide a holistic conceptual background for ICT adoption by SMEs. This integrative model comprises an overall taxonomy that categorizes some of the most internally and externally important elements influencing SMEs' ICT adoption. In a similar vein, [44] reviewed the literature on the link between ICT, SMEs, and poverty alleviation. First, this study examines the role of ICT adoption in SMEs. Then, it looks into how SMEs might adopt ICTs to help them mitigate poverty. The disparities in access and use of ICTs between firms have also been addressed in a review conducted by [45] on the digital divide (DD) among firms. This review considered the geographical area, the type of firm, and the time horizon of the study and the influence and causes of DD. Other studies reviewed the main determinants, effects, and barriers of ICT adoption in SMEs [27]. In addition, [46] analyzed the possible impacts of ICT on SMEs' performance. The importance of ICT skills and capabilities for SMEs has been targeted by [28], who discussed distinct perspectives on the topic. More recently, [26] also reviewed the subject of digital competencies in the workplace. To offer a first integrated perspective on organizational culture, sustainable development, and digitalization levels in SMEs, and their linkages, [47] proposed a conceptual model to show the relationships between these three topics. They concluded that the most studied cultural aspects were strategic alignment, organization core competencies, management, and viewpoints. Other recent studies reviewed digital innovation in SMEs [48] and concluded that it is guided by past experiences, progressing through many stages of innovation, culminating in organizational and business continuous improvement effects. In the same line of work, [49] presented a literature review to help distinguish the main difficulties and prospects faced by SMEs in the framework of digitalization and ICT advancements.

So far, none of the studies previously mentioned focused on the review of ICT policies. As highlighted by [50], there has been a lack of scholarly attention on how to select the best strategies for funding distribution across distinct ICT policies. In addition, just a few studies evaluate if this sort of funding distribution is made according to the most critical requirement of each region [51,52].

Since the effective strategy and application of suitable policy plans and programs involve familiarity with the main determinants regarding the ICT adoption and use by SMEs, we have conducted a literature review devoted to this topic, by using the keywords "ICT SMEs EU". For this purpose, the Google Scholar database was employed, only retaining the Q1 papers indexed in Scopus from 2012 onwards (representing the last 10 years). The results of the search are listed in Table 1.

**Table 1.** Studies on the assessment of the factors that influence ICT adoption in EU SMEs.

| Authors | Main Purpose | Methodologies | Variables |
|---|---|---|---|
| Ramdani et al. [53] | Study the technology, organization, and environment factors influencing SMEs' adoption of enterprise applications | Partial least squares technique | Dependent variable: adoption of enterprise applications. Independent variables: technological construct—relative advantage, compatibility, complexity, trialability, observability; organizational construct—top management support, organizational readiness, ICT experience, size; environmental construct—industry, market scope, competitive pressure, external ICT support |
| Hanclova et al. [54] | Study the main determinants of the adoption of ICT in micro and SMEs in Czech–Polish Border Areas | Asymmetric dependence testing and ordinal regression models | Explained variable—ICT adoption; determinants, ordinal data—data sources, modules in information systems, software properties, ICT maintenance |
| Billon et al. [55] | Investigate whether there are regional trends in ICT adoption in Europe and the role of geographical features in explaining household and firm ICT usage | Multivariate and canonical correlation analysis | Dependent variables: individuals using the Internet, individuals using a computer, individuals ordering goods and services online, employees using a computer at work, and employees using the Internet at work. Independent variables: economic variables—regional gross domestic product (GDP) per capita, employment specialization—employment in high- and medium high-technology manufacturing sectors, employment in total knowledge-intensive services, share of service employment over total, economic activity rate, research and development expenditure; human capital—life-long learning and tertiary education; ICT user characteristics—population between 15 and 64 years, population density, percentage of the population living in densely populated areas; institutional factors—the quality of government and the fiscal decentralization index |
| Chatzoglou and Chatzoudes [56] | Create and experimentally evaluate an analytical model for studying the main reasons influencing Greek SMEs' e-business use process | Exploratory factor analysis, confirmatory factor analysis, linear regression methods, and the structural equation modeling technique | Dependent variable: e-business adoption; independent variables: ICT infrastructure, Internet skills, firm size, firm scope, government support, consumer readiness |

**Table 1.** *Cont.*

| Authors | Main Purpose | Methodologies | Variables |
|---|---|---|---|
| Billon et al. [57] | Explore the presence of trends that integrate innovation and the usage of ICT by firms in the EU, as well as the reasons behind them | Factorial and cluster analyses | Dependent variables: Individuals using the Internet, individuals using a computer, individuals ordering goods and services online, employees using a computer at work, and employees using the Internet at work. Independent variables: economic variables—regional GDP per capita; employment specialization—employment in high and medium high-technology manufacturing sectors, employment in total knowledge-intensive services, share of service employment over total, economic activity rate, research and development expenditure; human capital—life-long learning and tertiary education; ICT user characteristics—population between 15 and 64 years, population density, percentage of the population living in densely populated areas; institutional factors—the quality of government and the fiscal decentralization index |
| Giotopoulos et al. [58] | Find possible determinants of ICT adoption in Greek SMEs | Ordered probit models | Dependent variables: ICT adoption—ICT intentions, ICT infrastructure, Internet integration, e-sales, e-procurement. Independent variables: technological competencies—organizational innovation, research and development (R&D) activities, research collaborations; human capital—personnel with scientific background, personnel with ICT skills; internal organization—decentralized decision-making, visionary leadership; environmental and firm characteristics—firm size, industry, and location |
| Ruiz-Rodríguez et al. [5] | Categorize and quantify DD in Spanish regions and those in EU MSs using data from firms with more than 10 workers that have adopted ICTs | Propose an enterprise digital development index (EDDI) for MSs in the EU and Spanish regions through factor analysis and cluster analysis | ICT connectivity to Internet—enterprises connected to the Internet, enterprises that employ ICT specialists; ICT adoption—enterprises with ERP package software, enterprises with homepage, use of social networks, workers using computers, enterprises with online publicity; e-commerce—enterprises using e-invoices; e-government—enterprises sending e-invoices to the government |
| Reggi and Gil-Garcia [50] | Examine the link between local demands and investment choices aimed at reducing regional DD in the EU | Ordinary least square models | Dependent variables: EU funding allocations to broadband, e-government, ICT in SMEs, and e-inclusion; independent variables: households with broadband availability, digital individuals who used the Internet for interaction with public authorities, enterprises sharing electronic information on the supply chain, individuals who have never used the Internet, Quality of Government Index, GDP per capita |

From the literature review conducted, it can be established that there is a scarcity of regional-level research and data on ICT at the firm level [5,50]. In addition, to the best of our knowledge, there are no studies that consider a fine-grained evaluation of the funds assigned to the OPs devoted to ICT adoption by SMEs in the programming period of 2014–2020.

Studies on the evaluation of ESIF devoted to ICT usually evaluate their impact, i.e., refer to ex post assessments [51,52]. In addition, there is a study that provides an ex ante evaluation in terms of the factors influencing the distribution of funding across distinct ICT strategies [50]. However, so far, there have been no studies that compare the implementation of the OPs devoted to ICT policies (regional or national) with their peers during the programming period, nor that identify the modifications that should occur to render an inefficient OP efficient. As a consequence, the adoption of nonparametric methodologies can be highly advantageous and appropriate, particularly since the indicators available for evaluating the Cohesion Policy can be used in conjunction with these types of methodologies.

Nonparametric techniques, including DEA, can easily handle many evaluation factors. In addition, DEA can assist in the identification of the primary factors that hinder efficiency, providing policymakers with pertinent information on how to overcome them. In this framework, [59] assessed the efficiency in the deployment of ESIF devoted to Objective 1 from 2000–2006 through a combination of the SFA and DEA techniques (the output-oriented version of the Charnes–Cooper–Rhodes (CCR) [60] and Banker–Charnes–Cooper (BCC) [61] models). Additionally, [62] applied the value-based DEA technique to appraise the execution of ESIF devoted to fostering the competitiveness of SMEs across different OPs (national and regional). In addition, [63] considered the output-oriented version of the slack-based model (SBM) combined with cluster analysis for assessing 102 OPs from 22 EU countries committed to fostering a low-carbon economy in SMEs. Finally, [64] employed the non-oriented version of the network SBM model coupled with cluster analysis to appraise 53 OPs from 19 countries dedicated to promoting research and innovation in SMEs.

Notwithstanding its utility, the DEA technique has yet to be used in the evaluation of ICT-related ESIF (see also Table A1 from Appendix A). The evaluation of efficiency through the DEA approach enables MAs to identify and follow the best practices, further understanding the necessary adjustments that must be operated on the set of metrics considered in the evaluation to attain efficiency.

Therefore, we will suggest a novel three-stage WRDD DEA model approach in conjunction with SFA, whose potentialities are then explored through its application in the evaluation of the implementation of OPs committed to boosting the use of ICT in the EU SMEs. With this novel methodology, further insights can be obtained that go beyond the use of traditional DEA methods, since it allows distinguishing those inefficiencies that result from management failures from those that result from potential environmental effects.

## 3. Methodology

We adopt the WRDD model which, contrarily to the CCR [60] and BCC [61] models, provides a more thorough analysis of efficiency since it is non-radial (inputs and outputs can vary non-proportionally) and can be non-oriented [65]. In addition, unlike radial models, the WRDD model provides information on the adjustments specifically required on every input and output to make each inefficient DMU efficient [65]. Finally, when compared to the SBM DEA model [66] it additionally provides the decomposition of inefficiency.

### 3.1. The Weighted Russell Directional Distance Model

Consider $n$ DMUs $(DMU_1, DMU_2, \ldots, DMU_n)$, and let the vectors of inputs and outputs of a given $DMU_o$ be given as $x_o$ and $y_o$, respectively. The WRDDM formulation proposed by [65] is then obtainable as:

$$
\begin{aligned}
&\max \beta_o^R = max \ (w_y(\textstyle\sum_r \omega_y^r \alpha_o^r) + w_x(\textstyle\sum_i \omega_x^i \zeta_o^i)) \\
&\text{s.t. } \textstyle\sum_{j=1}^n \lambda_j y_{rj} \ge y_{ro} + \alpha_o^r g_{yr}, \ r = 1, \ldots, s, \\
&\textstyle\sum_{j=1}^n \lambda_j x_{ij} \le x_{io} - \zeta_o^i g_{xi}, \ i = 1, \ldots, m, \\
&\textstyle\sum_{j=1}^n \lambda_j = 1, \ \lambda_j \ge 0, \ j = 1, \ldots, n,
\end{aligned}
\tag{1}
$$

where $\alpha_o^r$ and $\zeta_o^i$ are the inefficiency values for every output and input, respectively, and $\beta_o^R$ is the overall inefficiency (when $\beta_o^R = 0$, the DMU under evaluation is fully efficient); $\zeta_o^i g_{xi}$ refers to the reduction that should occur on the $i$-th input of $DMU_o$ to make it efficient. Similarly, $\alpha_o^r g_{yr}$ refers to the increase that should occur on the $r$-th output of $DMU_o$ to make it efficient. The weights $w_y$ and $w_x$ assign the importance related to the outputs and inputs, and $w_y + w_x = 1$. In addition, the importance of the inefficiencies associated with every output and input is defined such that $\sum_{r \in O} \omega_y^r = 1$, $\sum_{i \in I} \omega_x^i = 1$. In problem (1) it is considered that the directional vectors $g_x$ and $g_y$ are defined in such a way that $(-g_x, g_y) = (-x^o, y^o)$. Lastly, a variable returns to scale (VRS) technology is adopted with the imposition of $\sum_{j=1}^n \lambda_j = 1$, $\lambda_j \ge 0 \ (\forall_j)$.

Let $\alpha_o^{r*}$ and $\zeta_o^{i*}$ be the optimal solutions to problem (1). The following problem allows computing the reference set of each inefficient $DMU_o$:

$$
\begin{aligned}
&\max \ \textstyle\sum_r s_r^+ + \textstyle\sum_i s_i^- + \textstyle\sum_u s_u^-, \\
&\text{s.t. } \textstyle\sum_{j=1}^n \lambda_j y_{rj} - s_r^+ = y_{ro} + \alpha_o^{r*} g_{yr}, \ r = 1, \ldots, s, \\
&\textstyle\sum_{j=1}^n \lambda_j x_{ij} + s_i^- = x_{io} - \zeta_o^{i*} g_{xi}, \ i = 1, \ldots, m, \\
&\textstyle\sum_{j=1}^n \lambda_j = 1, \ \lambda_j \ge 0, \ j = 1, \ldots, n, \\
&s_r^+ \ge 0 \ (\forall_r), \ s_i^- \ge 0 \ (\forall_i)
\end{aligned}
\tag{2}
$$

Let $(s_r^{+*}, s_i^{-*}, \lambda_j^*)$ be the optimal solution to problem (2). On the one hand, the reference point of the efficient frontier for each inefficient $DMU_o$ is:

$$
(\hat{x}_o, \hat{y}_o) = (\textstyle\sum_{j \in E_o} \lambda_j^* x_j, \ \textstyle\sum_{j \in E_o} \lambda_j^* y_j)
\tag{3}
$$

On the other hand, the reference set of inefficient $DMU_o$ is:

$$
E_o = \left\{ j : \lambda_j^* > 0, j = 1, \ldots, n \right\}.
\tag{4}
$$

The WRDDM model's measure of inefficiency may be translated into an SBM departing from problem (5):

$$
\begin{aligned}
&\max \ (w_y(\textstyle\sum_r \omega_y^r \frac{s_r^{+\prime}}{g_{yr}}) + w_x(\textstyle\sum_i \omega_x^i \frac{s_i^{-\prime}}{g_{xi}})) \\
&\text{s.t. } \textstyle\sum_{j=1}^n \lambda_j y_{rj} = y_{ro} + s_r^{+\prime}, \ r = 1, \ldots, s, \\
&\textstyle\sum_{j=1}^n \lambda_j x_{ij} = x_{io} - s_i^{-\prime}, \ i = 1, \ldots, m, \\
&\textstyle\sum_{j=1}^n \lambda_j = 1, \ \lambda_j \ge 0, j = 1, \ldots, n, \\
&s_r^{+\prime} \ge 0 \ (\forall_r), \ s_i^{-\prime} \ge 0 \ (\forall_i).
\end{aligned}
\tag{5}
$$

Let $(s_r^{+*\prime}, s_i^{-*\prime}, \lambda_j^*)$ be the optimal solution to problem (5). The WRDDM inefficiency measure can be decomposed from:

$$
(w_y(\textstyle\sum_r \alpha_o^{r*\prime}) + w_x(\textstyle\sum_i \zeta_o^{i*\prime})), \text{ where } \alpha_o^{r*\prime} = \omega_y^r \frac{s_r^{+\prime}}{g_{yr}} \text{ and } \zeta_o^{i*\prime} = \omega_x^i \frac{s_i^{-\prime}}{g_{xi}}
\tag{6}
$$

### 3.2. Robustness Assessment

To cope with uncertainty, we assume as in [67] that the changes in the values of the factors are bounded by an interval, which is determined by utilizing a single tolerance value, $\omega$, such that $x_{ij}^L = x_{ij}(1-\omega) \le x_{ij} \le x_{ij}(1+\omega) = x_{ij}^U$ and $y_{ij}^L = y_{ij}(1-\omega) \le y_{ij} \le y_{ij}(1+\omega) = y_{ij}^U$.

The inputs and outputs are assumed to be bounded, respectively, by $[x_{ij}^L, x_{ij}^U]$ and $[y_{ij}^L, y_{ij}^U]$. Similarly, the corresponding directional vectors are also bounded by $[g_{xi}^L, g_{xi}^U]$ and $[g_{yr}^L, g_{yr}^U]$. In addition, in the present case, the weight profiles for all DMUs are held constant.

In the robustness assessment, worst-case and best-case scenarios are contemplated. In the first case, the outputs are increased while the inputs are decreased for all DMUs except for the DMU under evaluation (i.e., $DMU_o$ worsens its efficiency performance while the remaining DMUs improve their efficiency performance). In the second case, the opposite situation is assumed.

Problem (7) refers to the best-case scenario and allows obtaining the upper bound, $(1 - \beta_o^{LR})$, of the interval efficiency, $[(1-\beta_o^{UR}), (1-\beta_o^{LR})]$, for $DMU_o$:

$$
\begin{aligned}
&\max \beta_o^{LR} = max\ (w_y(\textstyle\sum_{r \in O} \varpi_y^r \alpha_o^r) + w_x(\textstyle\sum_{i \in I} \varpi_x^i \zeta_o^i)) \\
&\text{s.t. } \textstyle\sum_{j \ne o} \lambda_j y_{rj}^L \ge y_{ro}^U + \alpha_o^r g_{yr}^U,\ r = 1, \ldots, s, \\
&\textstyle\sum_{j \ne o} \lambda_j x_{ij}^U \le x_{io}^L - \zeta_o^i g_{xi}^L,\ i = 1, \ldots, m, \\
&\textstyle\sum_{j \ne o} \lambda_j z_{uj}^U \le z_{uo}^L,\ u = 1, \ldots, q, \\
&\textstyle\sum_{j \ne o} \lambda_j = 1,\ \lambda_j \ge 0,\ j = 1, \ldots, n.
\end{aligned}
\tag{7}
$$

Problem (8) refers to the worst-case scenario and allows obtaining the lower bound, $(1 -\beta_o^{UR})$, of the interval efficiency, $[(1 -\beta_o^{UR}), (1 -\beta_o^{LR})]$, for $DMU_o$:

$$
\begin{aligned}
&\max \beta_o^{UR} = max\ (w_y(\textstyle\sum_{r \in O} \varpi_y^r \alpha_o^r) + w_x(\textstyle\sum_{i \in I} \varpi_x^i \zeta_o^i)) \\
&\text{s.t. } \textstyle\sum_{j \ne o} \lambda_j y_{rj}^U \ge y_{ro}^L + \alpha_o^r g_{yr}^L,\ r = 1, \ldots, s, \\
&\textstyle\sum_{j \ne o} \lambda_j x_{ij}^L \le x_{io}^U - \zeta_o^i g_{xi}^U,\ i = 1, \ldots, m, \\
&\textstyle\sum_{j \ne o} \lambda_j z_{uj}^L \le z_{uo}^U,\ u = 1, \ldots, q, \\
&\textstyle\sum_{j \ne o} \lambda_j = 1,\ \lambda_j \ge 0,\ j = 1, \ldots, n.
\end{aligned}
\tag{8}
$$

From problems (7) and (8), it is possible to conclude that $1 - \beta_o^{UR} \le 1 - \beta_o^{LR}$.

Let J be the index set of DMUs ($j = 1, \ldots, n$). The DMUs can then be classified into strongly efficient, i.e., $E^{++} = \{j \in J: (1 - \beta_o^{UR}) \ge 1\}$; potentially efficient, i.e., $E^+ = \{j \in J: (1 - \beta_o^{UR}) < 1$ and $(1 - \beta_o^{LR}) \ge 1\}$; and strongly inefficient, i.e., $E^{--} = \{j \in J: (1 - \beta_o^{LR}) < 1\}$.

A DMU is robustly efficient (inefficient) to data perturbations if it stays efficient (inefficient) for the tolerance employed.

### 3.3. Stochastic Frontier Analysis

One of the limitations of the DEA approach is that it does not account for the impact of environmental factors and random errors in efficiency assessment. Hence, the single use of DEA can lead to unreasonable outcomes. To overcome this limitation, [68] proposed a three-stage DEA model. Firstly, the efficiency scores of every DMU and the input and output slacks are computed through the use of the DEA model. Secondly, the slacks are decomposed into environmental effects, managerial inefficiencies, and statistical noise. We use stochastic frontier analysis (SFA) [69,70] to implement the decomposition and, consequently, compute the adjusted input and output factors. The slacks are the dependent variables while the external environmental factors are the independent variables. The purpose is to remove the influence of environmental factors and random errors. The

functional form of the SFA model for each input slack obtained for *j* inefficient DMU ($j = 1, \ldots, p$) is given by:

$$s_{ij} = f\left(Z_j, \beta^i\right) + v_{ij} + u_{ij}, \;\; i = 1, \ldots, m; \; j = 1, \ldots, p \tag{9}$$

where $s_{ij}$ is the slack value of the *i*-th input of the *j*-th DMU, $f\left(Z_j, \beta^i\right)$ is the deterministic feasible slack frontier and $\beta^i$ designates the coefficients related to the environmental variables. The term $v_{ij} + u_{ij}$ is the mixed error, $v_{ij}$ refers to the statistical noise, and $u_{ij}$ refers to the management inefficiency. Generally, it is presumed that $v_{ij} \sim N\left(0; \sigma_v^2\right)$ and $u_{ij} \sim N^+\left(\mu^i; \sigma_u^2\right)$, with $v_{ij}$ and $u_{ij}$ being independent variables.

Let $\gamma = \frac{\sigma_u^2}{\sigma_u^2 + \sigma_v^2}$. If $\gamma$ is near 1, it means that management factors are in a leading position; i.e., most of the change required to attain efficiency is linked to management inefficiency. If $\gamma$ is near 0, the random error is the predominant factor; i.e., most of the change required to attain efficiency is related to statistical noise.

The adjusted input and output slacks are then computed by decomposing the mixed error. In line with [71], the management inefficiency is computed as follows:

$$E\left(u_{ij}\middle| u_{ij} + v_{ij}\right) = \frac{\sigma\delta}{1 + \delta^2}\left[\frac{\varphi\left(\frac{\varepsilon_j\delta}{\sigma}\right)}{\varnothing\left(\frac{\varepsilon_j\delta}{\sigma}\right)} + \frac{\varepsilon_j\delta}{\sigma}\right] \tag{10}$$

where $\delta = \frac{\sigma_u}{\sigma_v}$, $\varepsilon_j = v_{ij} + u_{ij,}$, $\sigma^2 = \sigma_u^2 + \sigma_v^2$, and $\varphi$ and $\varnothing$ correspond to the density function and distribution function of the standard normal distribution, respectively. Hence, the random error term can be computed as follows:

$$E\left(v_{ij}\middle| u_{ij} + v_{ij}\right) = s_{ij} - f\left(Z_j, \beta^i\right) - E\left(u_{ij}\middle| u_{ij} + v_{ij}\right) \tag{11}$$

We employ an adaption of the three-stage method of [68] by computing in the first stage the slacks through the WRDD DEA model, specifically through problem (5), instead of using the SBM DEA approach.

In stage 2, the input and output variables of each DMU are adjusted based on the SFA results by removing the significant environmental effects and statistical noises.

The input data are adjusted as follows [68]:

$$x_{ij}^A = x_{ij} + \left[\max_i\left\{f\left(Z_j, \beta^i\right)\right\} - f\left(Z_j, \beta^i\right)\right] + \left[\max_j\{v_{ij}\} - v_{ij}\right] \tag{12}$$

Considering a similar procedure for computing the adjusted outputs, these are adjusted as follows [72]:

$$y_{rj}^A = y_{rj} + \left[f\left(Z_j, \beta^r\right) - \min_r\{f\left(Z_j, \beta^r\right)\}\right] + \left[v_{rj} - \min_r\{v_{rj}\}\right] \tag{13}$$

Finally, in stage 3, we compute the efficiency scores with the adjusted input and output values.

## 4. Data and Assumptions

To account for the ICT SME support, we consider the dimensions of intervention given in Table 2, also used in the study of ICT funding allocations considered for SMEs by [50]. These two dimensions represent EUR 1.7 billion and EUR 304 million of planned investments, respectively [40]. These amounts are inscribed under multi-TO (EUR 810 million), TO2 (EUR 790 million), and TO3 (EUR 349 million) and to a smaller level under TO1 and TO8 [40].

**Table 2.** Dimensions of intervention considered.

| Code | Dimension |
|:---:|:---|
| 4 | Productive investment linked to the cooperation between large enterprises and SMEs for developing information and communication technology (ICT) products and services, e-commerce, and enhancing demand for ICT<br>Example of activities supported [40]: commerce and expand demand for ICT; encouragement and information initiatives (events, campaigns, consultation) aimed at increasing SME ICT preparedness, as well as the launch of innovative business ICT tools and solutions (ERP, CRM, cloud, among others) for SMEs. |
| 82 | ICT services and applications for SMEs (including e-commerce, e-business, and networked business processes), living labs, web entrepreneurs, and ICT start-ups). Example of activities supported [40]: Encourage the development of new advanced ICT solutions;<br>collaboration between ICT companies and academic institutions through incentive and communication campaigns, events, seminars, and business consulting; commercialization and global marketing of ICT goods and services (consultation, promotion, marketing, involvement in tenders and expos, software localization); aid SMEs in the development of innovative applications and services, such as smart linked devices; smart housing and energy efficiency; ICT services and applications for health, SMEs in the health sector, domotic services. |

We considered cumulative figures from previous years reported in March 2022. Furthermore, we only included in our assessment the OPs with no incomplete information (i.e., the OPs with incomplete information were not evaluated). This led to the assessment of 51 OPs from 16 countries.

We involved multiple stakeholders in the prior identification of the indicators employed in the evaluation by organizing a facilitated workshop with key policymakers and MAs on the subject "evaluating co-financed intervention policies in enterprises".

*4.1. Input and Output Factors*

The input and output factors selected for assessing the efficiency of the implementation of the ERDF funds committed to ICT support in SMEs were picked from a set of transversal indicators officially mandated by the EU, i.e., from the list of categorization data for the ERDF OPs (available online: https://cohesiondata.ec.europa.eu/2014-2020-Categorisation/ESIF-2014-2020-categorisation-ERDF-ESF-CF-planned-/3kkx-ekfq (accessed 30 March 2022)), and are further explained below.

4.1.1. The Rate of OPs' Execution

An efficient policy implementation implies an efficient financial execution of OPs, with a particular focus on the pace of execution of the OPs ([62–64,73]). In this context, costs must be certified by an approved authority, which is the entity in charge of certifying that the co-funded equipment/services were supplied, that the corresponding payments were made, and that the payments conformed with the EU and national rules. The indicators used to evaluate the capacity of the OPs' absorption are "total eligible spending" and "eligible costs decided". The first relates to the qualified costs reported and verified by this authority. Consequently, this indicator is utilized as an output because the greater the amount assigned to it, the greater the financial implementation of every project. The second is regarded as an input since it refers to the financial funds devoted to the projects selected for funding, and this should be minimized. Since the assignment of the ICT funds to distinct policy objectives is influenced by the plans that policymakers agree to follow [50], this latter indicator has also been used in the second stage of the analysis to understand how it was affected by ICT strategies in SMEs.

### 4.1.2. Operations Supported

The "number of operations supported" relates to the number of projects that obtained ERDF funding. The greater the number of supported projects, the greater the prospect of increasing/enhancing the firms' use of ICT. Hence, this indicator is viewed as an output.

Data on these indicators are available in Table A2 in Appendix A.

Based on Table 3, it is reasonable to infer that the overall average financial implementation ratio (i.e., the ratio of "total eligible expenditure" and "total eligible cost") is somewhat higher than 50% (56.3%). Furthermore, the number of projects financed varies greatly.

**Table 3.** Descriptive statistics of the inputs and outputs.

| Statistics | Total Eligible Spending | Number of Operations | Total Eligible Costs Decided |
|---|---|---|---|
| Mean | 15,861,300 | 409 | 28,169,468 |
| Median | 3,238,795 | 27 | 5,000,000 |
| Standard Deviation | 38,520,025 | 1068 | 63,497,428 |
| Minimum | 68,486 | 1 | 251,294 |
| Maximum | 237,904,467 | 5457 | 311,154,920 |
| Count | 51 | 51 | 51 |

Source: Data available online at https://cohesiondata.ec.europa.eu/2014-2020-Categorisation/ESIF-2014-2020-categorisation-ERDF-ESF-CF-planned-/3kkx-ekfq (accessed on 30 March 2022).

A prerequisite of DEA is that inputs and outputs should hold an isotonic relationship [74]. This property indicates that raising any input whilst holding the other evaluation factors constant must not lower any output but rather raise at least some output. A positive and significant correlation between inputs and outputs depicts an isotonic relationship. We opted to compute the Spearman correlation coefficients and the related significance tests since the assumption of normality for the applicability of the tests for the significance of Pearson's correlation was not confirmed in this case—see Table 4.

**Table 4.** Correlation matrix.

| Variable | Operations Supported | Eligible Cost Decided | Total Eligible Spending |
|---|---|---|---|
| Operations supported | 1 | | |
| Eligible cost decided | 0.71 ** | 1 | |
| Total eligible spending | 0.77 ** | 0.91 ** | 1 |

Note: ** Significant at 1% level.

### 4.2. Environmental Factors

Because of the absence of data on ICT adoption at the firm level from traditional data sources at the NUTS2 level [50,55,57], we ended up using, in the second stage of the analysis, indicators available from the Regional Innovation Scoreboard in 2021 [75]. The indicators were selected according to the literature review on the main determinants that influence ICT adoption in EU SMEs (see Table 1).

Concerning independent variables (see Table 5), we have employed as a proxy of economic development the regional GDP at purchasing power parity per capita (GDPPPpc) [50,55,57]. As documented by [76], ICT tends to be promoted more efficiently in prosperous regions. Additionally, since ICT goods and services are classified as normal goods [77], their demand is also supposed to vary positively with GDP.

**Table 5.** Descriptive statistics of the environmental factors.

| Environmental Factors | Mean | Standard Deviation | Min | Max |
|---|---|---|---|---|
| Population with tertiary education | 0.5767 | 0.1916 | 0.1156 | 1 |
| Digital skills | 0.5359 | 0.1949 | 0.2814 | 0.9318 |
| R&D expenditures business sector | 0.3105 | 0.2101 | 0.0215 | 0.8024 |
| ICT specialists | 0.4018 | 0.2527 | 0.0470 | 1 |
| Product process innovators | 0.5529 | 0.2511 | 0.1767 | 1 |
| GDPPPP$_{pc}$ | 87.72 | 23.9315 | 49.09 | 178.30 |

We have also considered as an environmental factor the percentage population aged 25–34 having completed tertiary education, because the literature acknowledges a positive linkage between educational achievement and ICT adoption [55,57]. Particular reasons have been stressed in the case of ICT use to explain this possible positive association. On the one hand, education offers the skills needed to use and benefit from ICT. On the other hand, employees should be expected to be more competent at learning how to utilize new technologies, specifically ICT [57].

Additionally, since R&D expenditures foster ICT dissemination for European regions [55,57,58], we have used as an explanatory variable R&D expenditure in the business sector as a percentage of GDP and SMEs introducing product innovations as a percentage share of all SMEs.

In addition, because the ICT skills of a firm are regarded as important technology-related elements that are influential in determining user acceptance and ICT deployment [58], we have considered individuals who have above basic overall digital skills as a percentage of total SME employment. Finally, the percentage of ICT specialists as a percentage of total SME employment has similarly been used, i.e., people whose primary occupation is ICT and who are capable of handling a wide variety of responsibilities connected to firms' computer systems [5].

Data were normalized using the min–max procedure, i.e., the minimum score for all regions across all the available years was subtracted from the respective transformed score, which was then divided by the difference between the maximum and minimum scores observed for all data (regions and years). The maximum normalized score is equal to 1 and the minimum normalized score is equal to 0—see Tables 5 and A3.

## 5. Analysis and Discussion of Results

Results were computed through an Excel Visual Basic based application that was specifically developed by the authors for solving our DEA problems and uses Excel Solver as the backend. Table 6 displays the basic descriptive statistics of the results obtained for the OPs under evaluation.

**Table 6.** Descriptive statistics of the results obtained both for efficient and inefficient OPs.

| | Statistics | Efficiency | Total Eligible Spending | Number of Operations | Total Eligible Costs Decided |
|---|---|---|---|---|---|
| Efficient DMUs | Mean | 1.09 | 46,026,233.00 | 1310.40 | 75,514,839.90 |
| | Median | 1.07 | 9,217,730.00 | 339.50 | 9,633,113.00 |
| | Standard Deviation | 0.10 | 74,818,282.11 | 2108.72 | 118,719,405.31 |
| | Minimum | 1.00 | 329,249.00 | 1.00 | 251,294.00 |
| | Maximum | 1.31 | 237,904,467.00 | 5457.00 | 311,154,920.00 |
| | Count | 10 | 10 | 10 | 10 |
| Inefficient DMUs | Mean | −17.54 | 8,503,999.66 | 189.54 | 16,621,815.98 |
| | Median | −6.21 | 1,963,414.00 | 14.00 | 4,901,930.00 |
| | Standard Deviation | 35.48 | 17,671,362.04 | 415.58 | 34,228,777.84 |
| | Minimum | −205.37 | 68,486.00 | 1.00 | 373,794.00 |
| | Maximum | 0.98 | 102,175,668.00 | 2184.00 | 202,847,237.00 |
| | Count | 41 | 41 | 41 | 41 |

Based on Table 6, it is fair to assert that, while efficient OPs have significantly higher average scores than inefficient ones (with efficiency values ranging between 1.00 and 1.31 and with at least 50% of efficient OPs with efficiency values higher than 1.07), their evaluation factors are also more variable when compared to those of inefficient OPs (these findings are corroborated by the higher standard deviation values). Furthermore, inefficient OPs have a broad range of efficiency ratings (with a standard deviation of 35.48 and at least 50% of inefficient OPs with efficiency values lower than −6.61).

The number of OPs at distinct efficiency score subintervals is illustrated in Figure 1.

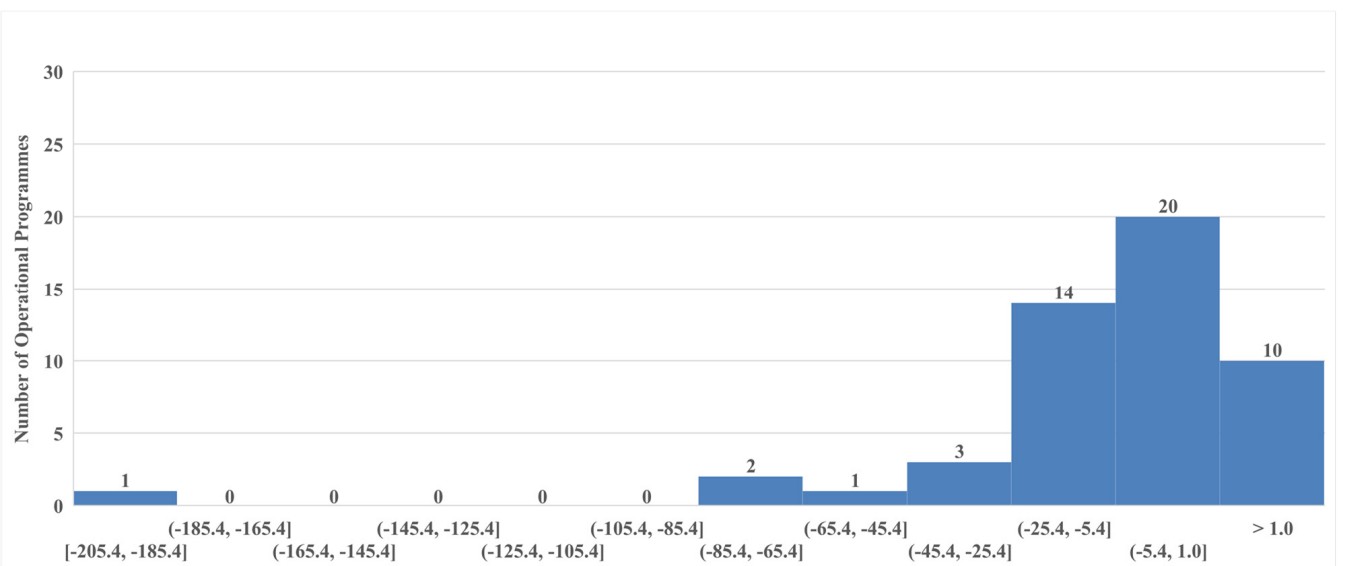

**Figure 1.** Number of OPs at distinct efficiency score subintervals.

Only about 20% of the OPs attained an efficient procedural performance, i.e., 10 out of 51 (Figure 1).

The four OPs that are most frequently selected as benchmarks are "Extremadura—ERDF" (30), "País Vasco—ERDF" (25), "Multi-regional Spain—ERDF" (22) and "Provence-Alpes-Côte d'Azur—ERDF/ESF/YEI" (21)—see Table 7. Curiously, all these regions are located in countries viewed as being the highest spenders on ICT support for SMEs [38]. Furthermore, the findings obtained for Spanish regional OPs (the country with the OPs most frequently selected as benchmarks) match the conclusions of [5]. In line with the authors of [5], Spanish regions at the firm level are at a moderate level of digital advancement or at a greater level of digital advancement than their EU peers, as well as displaying a reduced DD among them (i.e., a smaller degree of difference in firm digital development) in comparison to what occurs in the remaining European MSs. Curiously, Greece and Bulgaria, countries with OPs classified as efficient, were identified in [5] as countries whose firms were last ranked in terms of digital inclusion (according to 2015 data), highlighting the efforts made by these countries in the adoption of ICT by SMEs in the latest programming period.

Figure 2 demonstrates that the average funds committed to the eligible cost of efficient OPs (EUR 75,514,840) were much higher than those dedicated to inefficient OPs (EUR 16,621,816). Similar conclusions may be drawn for the mean eligible spending of efficient OPs (EUR 46,026,233) and inefficient ones (EUR 8,504,000)—see Figure 2. Furthermore, the number of operations enabled by efficient OPs is substantially greater than that of inefficient OPs (1310 against 190).

**Table 7.** Main characteristics of efficient OPs.

| MS (2 Digit ISO) | OP | No. of Times as Benchmark | Rank | Total Eligible Spending | Number of Operations | Total Eligible Costs Decided |
|---|---|---|---|---|---|---|
| FR | Provence-Alpes-Côte d'Azur—ERDF/ESF/YEI | 21 | 1 | 329,249 | 1 | 251,294 |
| ES | Extremadura—ERDF | 30 | 2 | 1,560,112 | 810 | 4,823,735 |
| CZ | Enterprise and Innovation for Competitiveness—CZ—ERDF | 2 | 3 | 237,904,467 | 451 | 311,154,920 |
| ES | Multi-regional Spain—ERDF | 22 | 4 | 58,864,158 | 5108 | 95,971,219 |
| ES | País Vasco—ERDF | 25 | 5 | 3,964,897 | 575 | 4,618,616 |
| GR | Competitiveness Entrepreneurship and Innovation—GR—ERDF/ESF | 1 | 6 | 100,667,978 | 5457 | 275,856,182 |
| BG | Innovations and Competitiveness—BG—ERDF | 6 | 7 | 33,942,154 | 228 | 38,612,352 |
| LT | EU Structural Funds Investments—LT—ERDF/ESF/CF/YEI | 1 | 8 | 7,607,793 | 210 | 7,786,656 |
| GR | Epirus—ERDF/ESF | 4 | 9 | 4,593,855 | 144 | 4,593,855 |
| PL | Podkarpackie Voivodeship—ERDF/ESF | 1 | 10 | 10,827,667 | 120 | 11,479,570 |

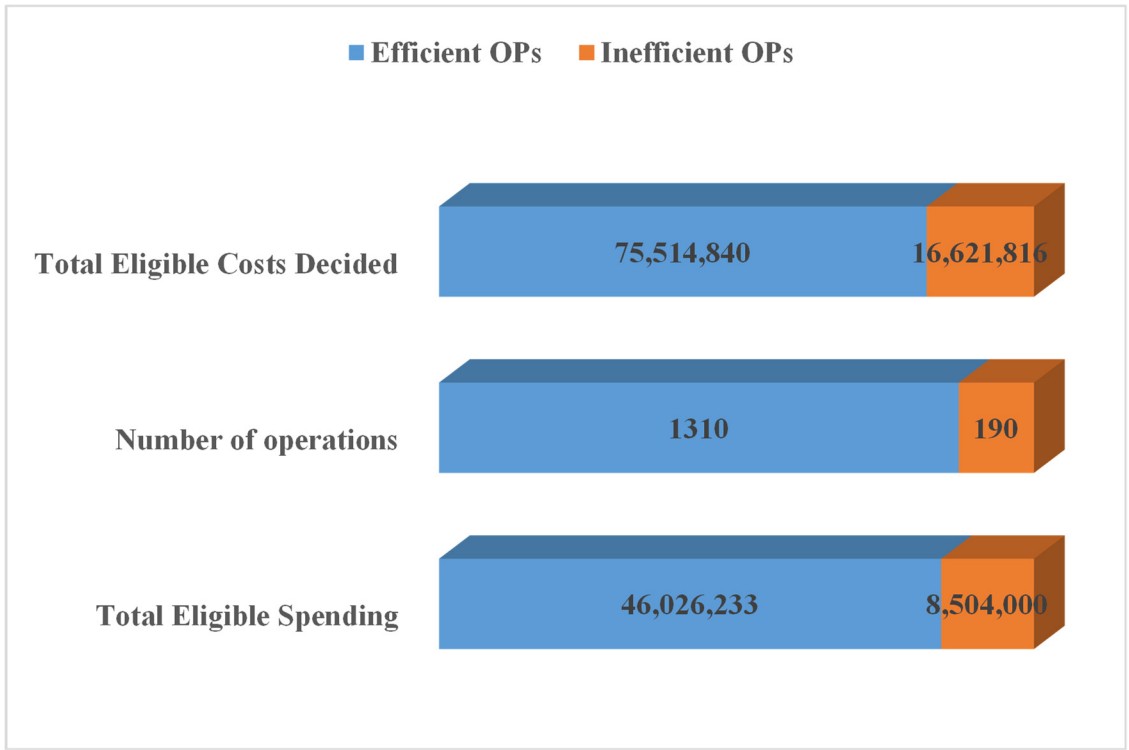

**Figure 2.** Average input and output values attained for the efficient and inefficient OPs.

Figure 3 shows the contribution of each factor to the inefficiency recorded in inefficient OPs. Our findings indicate that the number of operations supported is the factor that requires more attention from MAs. The OPs' financial execution is also important but in fewer cases. While nine OPs require an enhancement of "eligible spending", there is only one OP that requires the reduction in funding dedicated to it, i.e., the reduction in "eligible costs decided".

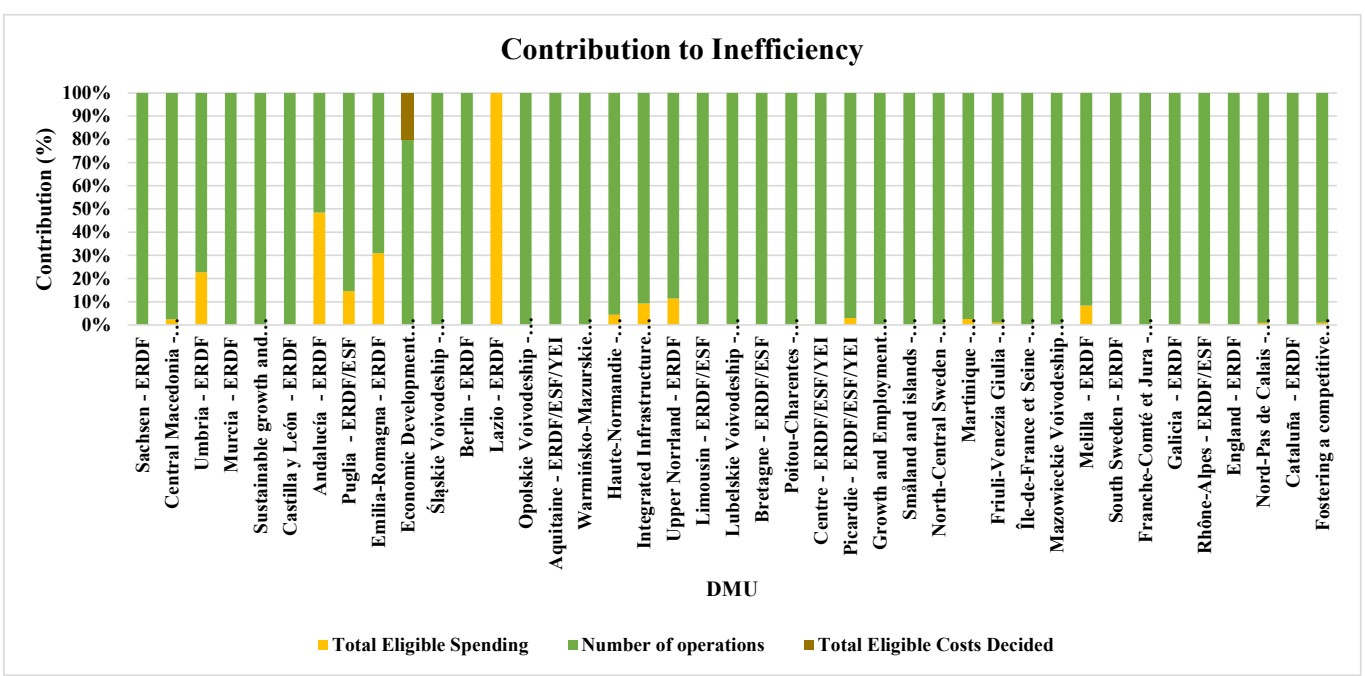

**Figure 3.** Contribution of inputs and outputs to inefficiency.

The WRDD DEA model also provides information on the adjustments that inputs and outputs must undergo to make inefficient OPs efficient (Table 8). These results are also displayed in Figures 4 and 5 per OP, where OPs are portrayed in ascending order of efficiency from left to right.

**Table 8.** Improvement potential for the OPs.

| Factor | Average Original | Average Projection | Variation |
|---|---|---|---|
| Total Eligible Spending | 8,504,000 | 9,912,457 | 17% |
| Number of Operations | 189.54 | 902.34 | 376% |
| Total Eligible Costs Decided | 16,621,816 | 15,284,716.33 | −8% |

The "number of operations supported" shows the largest improvement potential of about 376% (i.e., it should increase on average from 189.54 to 902.34), while "eligible costs decided" (−8%) and "eligible spending" (17%) only require mild adjustments—see Table 8 and Figures 4 and 5.

*5.1. Robustness Analysis*

The robustness study was carried out using tolerances of 5% and 10% for introducing data perturbation in all factors under scrutiny. According to our findings shown in Figure 6, it can be concluded that there are only four efficient OPs that remain efficient for data perturbations with both tolerances, which are "Provence-Alpes-Côte d'Azur—ERDF/ESF/YEI", "Extremadura—ERDF", "Enterprise and Innovation for Competitiveness—CZ—ERDF", and "Multi-regional Spain—ERDF". Contrastingly, 21 OPs remain robustly inefficient for both data perturbations (i.e., 41% of the sample of OPs under evaluation). Finally, under the same conditions, 27 OPs are potentially efficient. All in all, these outcomes suggest a poor application of ERDF funds in ICT SME support. These findings appear to be consistent with the challenges identified by [38] at the heart of the EU ICT policy in terms of digital achievements. According to their findings, the EU seems to underachieve its competitors (specifically, the United States, Japan, and South Korea) in terms of innovative ICT infrastructures and ICT adoption, in particular by SMEs. In fact, [33] ascertained that to create the encouraging circumstances and provide incentives to SMEs for adopting innovative

solutions, SMEs need explicit policy initiatives to guarantee that technological services can be supplied and that the required infrastructures are accessible.

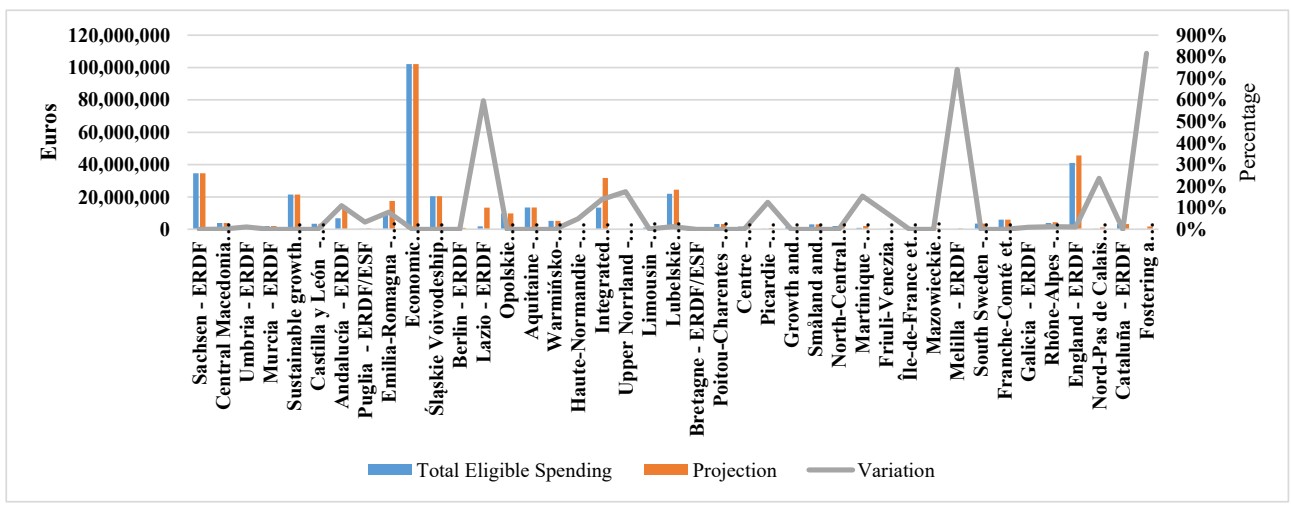

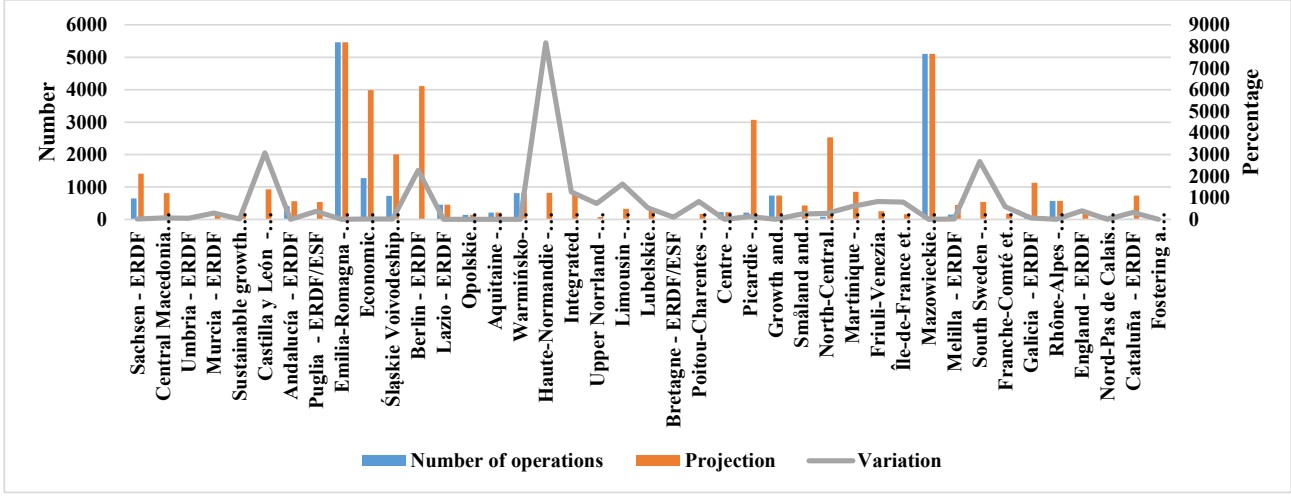

**Figure 4.** Potential output improvements for every inefficient OP.

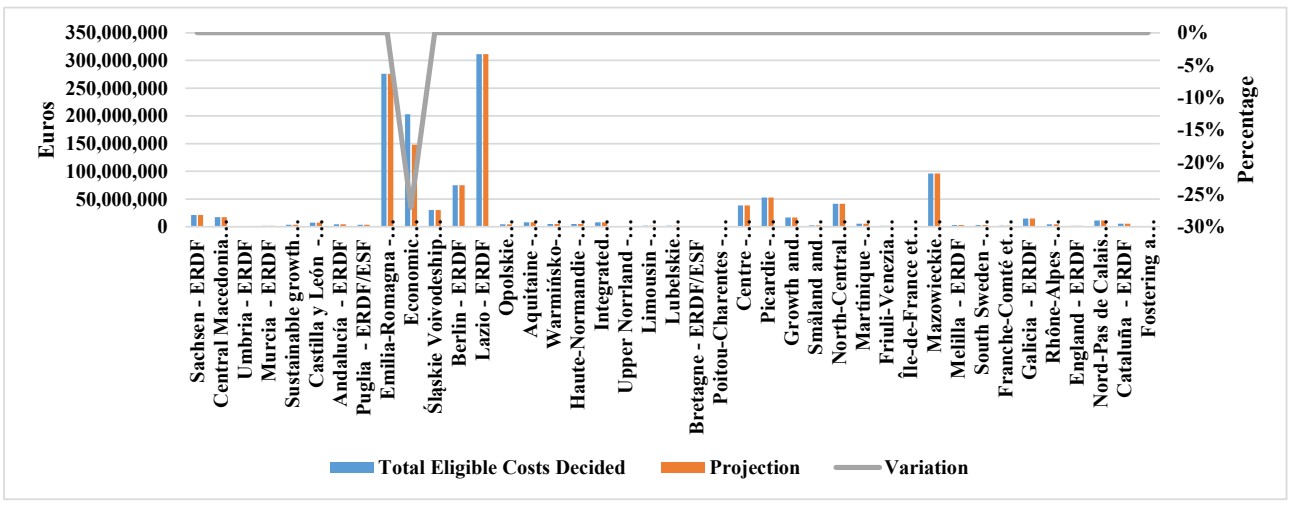

**Figure 5.** Potential input improvement for every inefficient OP.

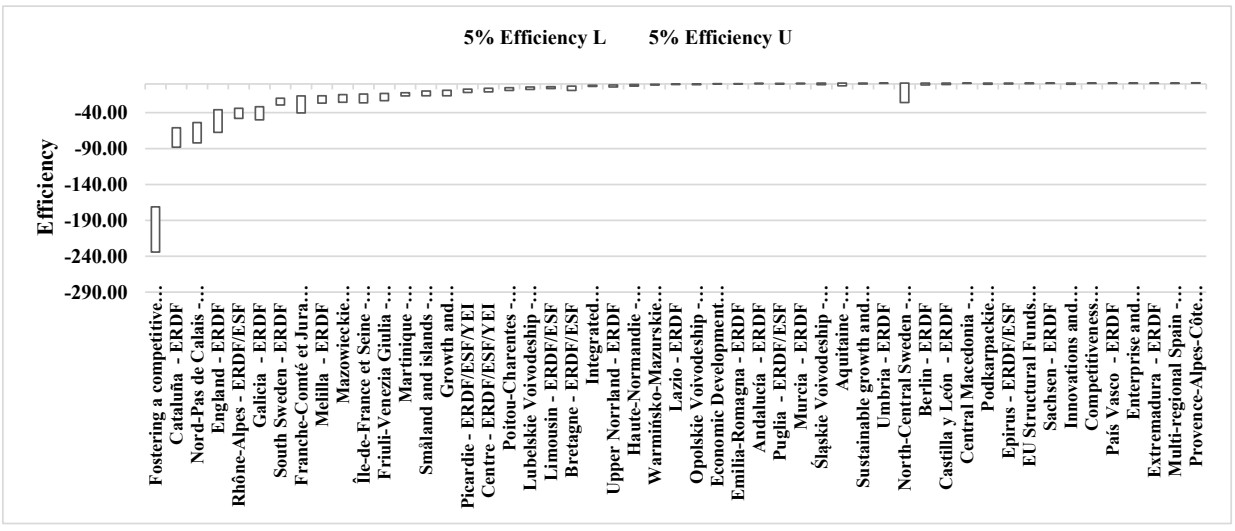

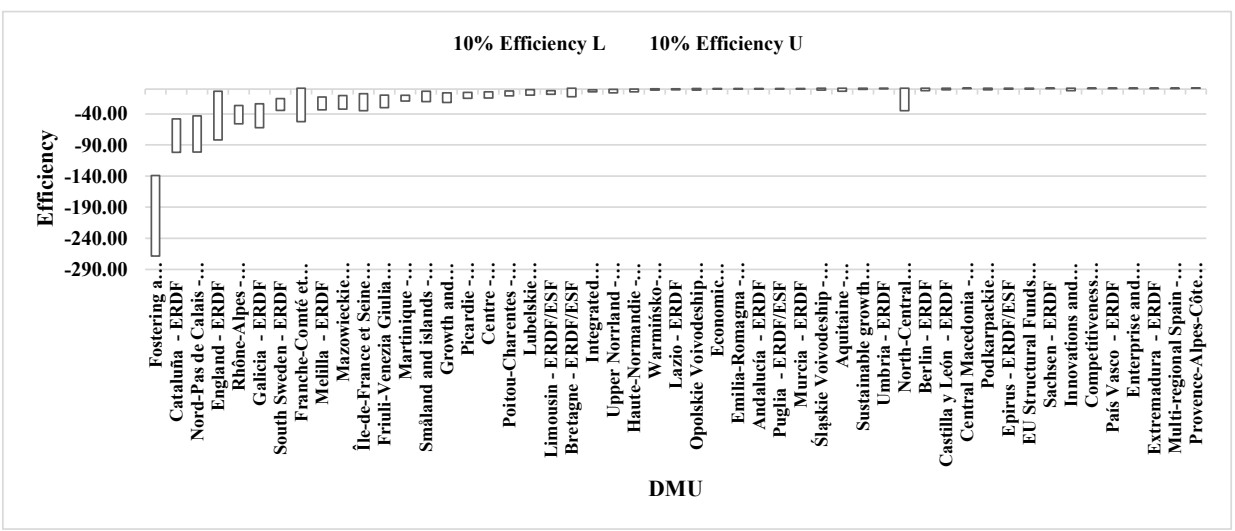

**Figure 6.** Results of robustness analysis per OP.

### 5.2. Analysis of SFA Regression Results

The slacks of output factors obtained with problem (5) are used as the explained variables at this stage, resulting in two regression models (since there was only one OP with an input surplus, these factors were not adjusted). In this context, correlation coefficients and the variance inflation factors (VIFs) are two different methods for examining multicollinearity among the variables. A high correlation between two explanatory variables is an indicator of multicollinearity. Table 9 presents the correlation analysis between the environmental variables.

It can be seen that there is a strong correlation (0.73) between GDPPPPPC and ICT specialists. Otherwise, all the other significant correlations correspond to moderate correlations since the coefficients vary between 0.32 and 0.65. Since multicollinearity is defined as a phenomenon in which two or more explanatory variables in a multiple regression model are highly correlated, the values of VIFs were also calculated (see Table 10). The higher the VIF is, the more likely the multicollinearity among the variables is. VIF starts at 1 and has no upper limit, and values between 1 and 5 indicate a moderate correlation between two explanatory variables, but, within this range, it is considered not severe enough to require attention [78,79].

**Table 9.** Correlation coefficients.

| Variables | Population with Tertiary Education | Digital Skills | R&D Expenditures Business Sector | ICT Specialists | Product Process Innovators | GDP per Capita |
|---|---|---|---|---|---|---|
| Population with tertiary education | 1.00 | 0.32 * | 0.47 ** | 0.35 * | 0.16 | 0.44 ** |
| Digital skills | 0.32 * | 1.00 | 0.40 ** | 0.40 ** | 0.32 * | 0.48 ** |
| R&D expenditures business sector | 0.47 ** | 0.40 ** | 1.00 | 0.54 ** | 0.56 ** | 0.65 ** |
| ICT specialists | 0.35 * | 0.40 ** | 0.54 ** | 1.00 | 0.46 ** | 0.73 ** |
| Product process innovators | 0.16 | 0.32 * | 0.56 ** | 0.46 ** | 1.00 | 0.60 ** |
| GDPPPP$_{PC}$ | 0.44 ** | 0.48 ** | 0.65 ** | 0.73 ** | 0.60 ** | 1.00 |

Note: Pearson's correlation coefficients are presented in this table. The significance levels of ** and * are 1% and 5%, respectively.

**Table 10.** Variance inflation factor.

| Variables | VIF |
|---|---|
| Population with tertiary education | 1.4767 |
| Digital skills | 2.2855 |
| R&D expenditures business sector | 2.5844 |
| ICT specialists | 2.6999 |
| Product process innovators | 2.0949 |
| GDPPPP$_{PC}$ | 3.2690 |

The obtained results for multicollinearity indicate that we can include all variables in the initial models. However, due to the problem of misspecification, the final models do not necessarily contain all the variables previously considered.

The R program version 4.0.5 (RStudio Team, 2021) and, specifically, the sfaR package version 0.1.1 were used to run the SFA regression models [80]. Table 11 displays the results.

**Table 11.** Results obtained with SFA regression models.

| Variables | Slacks | |
|---|---|---|
| | Total Eligible Spending | Number of Operations |
| Constant | −696,940 ** | 248.61 ** |
| Population with tertiary education | −4,007,800 ** | - |
| Digital skills | −3,279,600 ** | 205.16 ** |
| ICT specialists | 6,634,300 ** | 143.90 ** |
| Product process innovators | −3,805,600 ** | −83.38 ** |
| GDPPPP$_{pc}$ | 15,499 ** | −3.52 ** |
| Sigma-squared | $2.85 \times 10^{13}$ ** | $1.08 \times 10^{6}$ ** |
| Gamma | 0.97 * | 0.99 * |
| Log-likelihood function | −672.65 | −331.54 |

* and ** Significance at the 5% and 1% levels, respectively.

The $\gamma$ values of the two models are close to 1 and significant at the 1% level, indicating that management factors are the main factors responsible for the efficiency scores attained. Therefore, to obtain unbiased efficiency results, we removed the effects of the environmental variables and random errors through the use of SFA.

All the regression coefficients are significant at the 1% level, suggesting that the selected environmental variables have a significant impact on the slacks of each output.

From Table 11, it can be concluded that the increases in both the percentage of ICT specialists and GDPPPP$_{pc}$ contribute to a higher required improvement of "total eligi-

ble spending", whereas the remaining factors have a negative influence on the required improvement of this factor. These findings suggest that more developed regions and a higher rate of ICT specialists do not necessarily mean a higher absorption rate of ERDF funds dedicated to boosting ICT in SMEs. The underuse of ERDF by ICT Croatian SMEs was also acknowledged by the authors of [81] for the period of 2014–2020. These authors concluded that the intricacy and the required time to apply, develop, and appraise the projects were the potential cause of these findings [81]. In addition, the authors of [82] ascertained that SMEs' investment ends up being smaller than it would be anticipated by standard economic fundaments, indicating that these firms are particularly vulnerable to funding issues. Another reason for these findings might reflect the use of other sources of financing by these SMEs [38].

Concerning the required increase in the "number of operations" supported, the results show that this variable tends to increase as the digital skills and ICT specialists increase, whereas it tends to decrease with the percentage of SMEs with product process innovations and $GDPPPP_{pc}$. Then again (and for similar previous reasons), these outcomes suggest that a higher rate of ICT skills/specialists does not necessarily lead to an efficient number of operations supported. On the other hand, more developed regions and a higher percentage of SMEs more prone to process innovations do not necessarily need to run for additional ERDF-supported operations, because these are more efficient in applying for funding.

*5.3. Efficiency after Input and Output Adjustments*

In the first stage of the analysis, we found that the mean efficiency scores computed were quite volatile, validating the use of the multistage technique [83]. In the second stage, we used the SFA approach to adjust the outputs by removing the effects of environmental factors and random shocks. Subsequently, we obtained the inherent production possibility set by running problem (1) with these new adjusted factors.

According to Table 12, it is possible to conclude that efficient OPs reduce their variability, with a standard deviation of 0.05 (efficiency values vary within [1.00, 1.18] and with more than 50% of efficient OPs with efficiency values higher than 1.03). Moreover, inefficient OPs significantly reduce the variability of their efficiency ratings (with a standard deviation of 0.53 and with more than 50% of inefficient OPs with efficiency values lower than 0.51) and improve their performance substantially (underlining the importance of the impact of environmental factors).

**Table 12.** Descriptive statistics of the results obtained both for efficient and inefficient OPs with adjusted factors.

| | Statistics | Efficiency | Total Eligible Spending | Number of Operations | Total Eligible Costs Decided |
|---|---|---|---|---|---|
| Efficient DMUs | Mean | 1.05 | 31,428,303.50 | 1084.51 | 48,760,091.13 |
| | Median | 1.03 | 5,110,867.00 | 334.43 | 1,394,419.50 |
| | Standard Deviation | 0.05 | 61,503,572.72 | 1736.75 | 99,207,159.68 |
| | Minimum | 1.00 | 329,249.00 | 1.00 | 251,294.00 |
| | Maximum | 1.18 | 237,904,467.00 | 5457.00 | 311,154,920.00 |
| | Count | 16 | 16 | 16 | 16 |
| Inefficient DMUs | Mean | 0.42 | 12,630,063.32 | 387.06 | 18,756,611.31 |
| | Median | 0.51 | 5,906,844.00 | 270.86 | 7,401,115.00 |
| | Standard Deviation | 0.53 | 19,135,345.69 | 287.45 | 36,006,785.23 |
| | Minimum | −1.89 | 2,426,946.00 | 120.00 | 1,502,834.00 |
| | Maximum | 0.89 | 108,321,040.00 | 1562.85 | 202,847,237.00 |
| | Count | 35 | 35 | 35 | 35 |

About 30% of the OPs reached an efficient procedural performance against the previous 20%, i.e., 16 out of 51 (Tables 6 and 12).

The difference between the efficiency scores of the OPs with and without adjusted factors is depicted in Figures 7 and 8. From there, it can be concluded that the efficiency scores obtained without an adjustment of the factors used in the analysis were underrated.

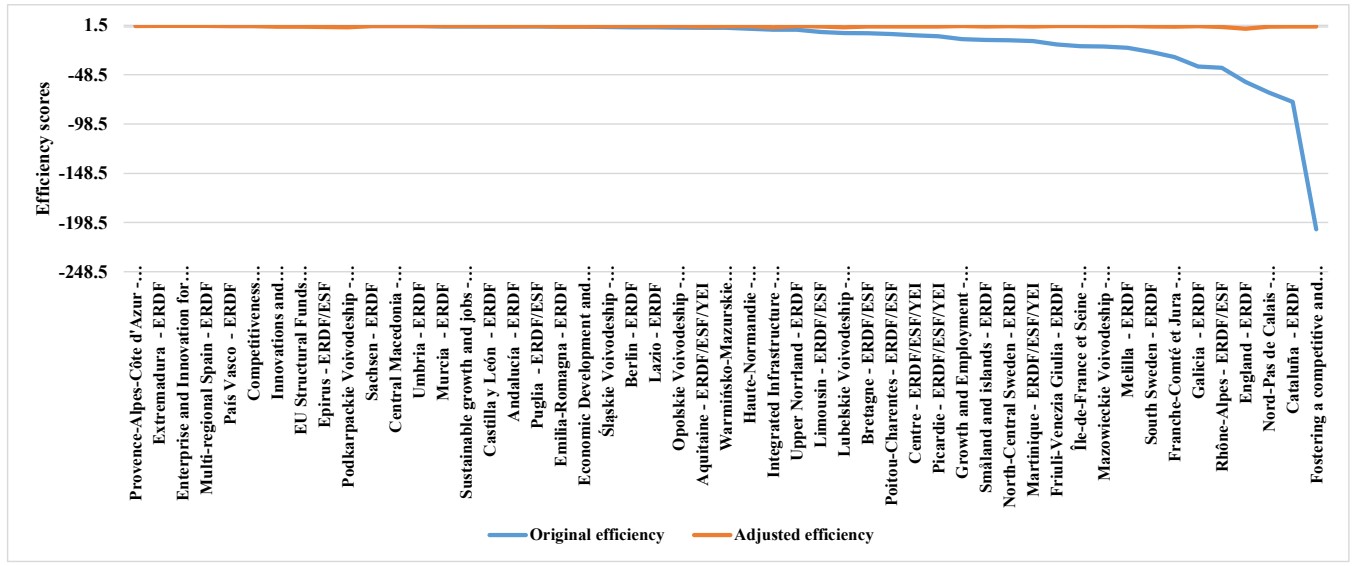

**Figure 7.** Original vs. adjusted efficiency scores.

When contrasted with the first stage, "Melilla—ERDF", "Île-de-France et Seine—ESF/ERDF/YEI", and "Friuli-Venezia Giulia—ERDF" showed the largest increase in efficiency, with their values rising from −21.05 to 1.02 (105%), −19.60 to 1.03 (105%), and −17.78 to 1.02 (84%), respectively—see Figure 8 and Table 12. These findings suggest that these OPs' previous inefficiencies were not solely because of their backward technological level, but rather associated with their external environment.

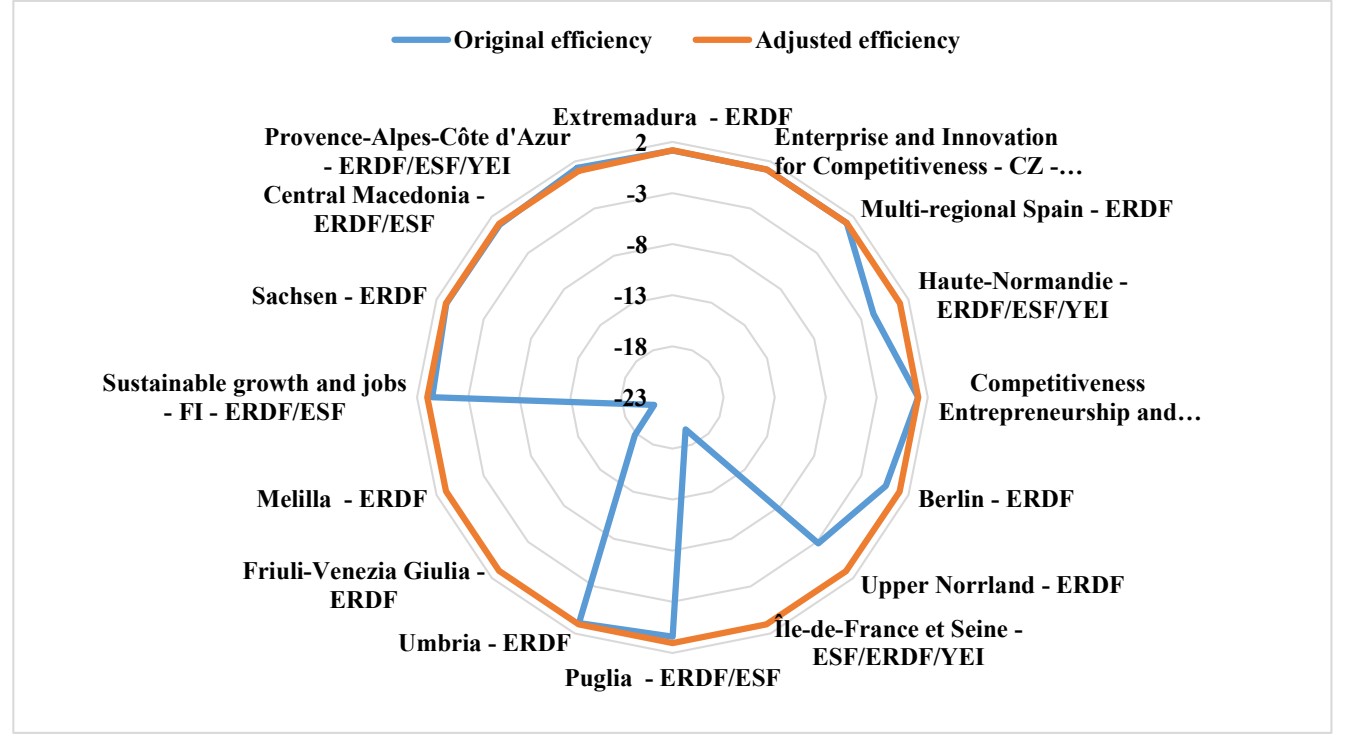

**Figure 8.** Original vs. adjusted efficiency scores for the 16 efficient OPs obtained with adjusted factors.

The OPs that are more often selected as benchmarks are "Umbria—ERDF" (26), "Multi-regional Spain—ERDF" (19), "Central Macedonia—ERDF/ESF" (13), and "Berlin—ERDF" (11)—see Table 13. Curiously, "Multi-regional Spain—ERDF" remains in the top four efficient countries viewed as benchmarks. In addition, two (out of five) of the OPs that remain efficient regardless of the adjustments belong to Spanish regions (see grey cells of Table 12). In this context, it is worth mentioning that MSs in Southern (such as Italy and Spain) and Central and Eastern Europe were the biggest recipients of financing for ICT and the digital economy [38]. This is particularly true for countries with efficient OPs, such as Spain, Greece, Hungary, the Czech Republic, and the Baltic States [38]. Curiously, it is interesting to see that in some of the countries that significantly worsened their performance because of environmental factors, such as Poland (previously ranked 10th and dropped to 50th) and Bulgaria (previously ranked 7th and dropped to 43rd), more than 50% of their firms had extremely modest degrees of digitization in 2016 [38]. Finally, other OPs from countries such as Italy, Germany, and Finland also show the importance of environmental factors, revealing that with the adjusted outputs they become efficient.

**Table 13.** Main characteristics of efficient OPs obtained with and without the adjusted factors.

| MS (2 Digit ISO) | OP | Efficiency Score (without Adjusted Factors) | Efficiency Score (with Adjusted Factors) | No. of Times as Benchmark (without Adjusted Factors) | No. of Times as Benchmark (with Adjusted Factors) | Rank (without Adjusted Factors) | Rank (with Adjusted Factors) |
|---|---|---|---|---|---|---|---|
| FR | Provence-Alpes-Côte d'Azur—ERDF/ESF/YEI | 1.31 | 1.00 | 21 | 1 | 1 | 16 |
| ES | Extremadura—ERDF | 1.15 | 1.18 | 30 | 6 | 2 | 1 |
| CZ | Enterprise and Innovation for Competitiveness—CZ—ERDF | 1.14 | 1.14 | 2 | 3 | 3 | 2 |
| ES | Multi-regional Spain—ERDF | 1.13 | 1.12 | 22 | 19 | 4 | 3 |
| ES | País Vasco—ERDF | 1.07 | 0.88 | 25 | - | 5 | 19 |
| GR | Competitiveness Entrepreneurship and Innovation—GR—ERDF/ESF | 1.07 | 1.07 | 1 | 1 | 6 | 5 |
| BG | Innovations and Competitiveness—BG—ERDF | 1.02 | 0.40 | 6 | - | 7 | 43 |
| LT | EU Structural Funds Investments—LT—ERDF/ESF/CF/YEI | 1.01 | 0.39 | 1 | - | 8 | 44 |
| GR | Epirus—ERDF/ESF | 1.01 | 0.15 | 4 | - | 9 | 46 |
| PL | Podkarpackie Voivodeship—ERDF/ESF | 1.00 | −0.51 | 1 | - | 10 | 50 |
| FR | Haute-Normandie—ERDF/ESF/YEI | −1.71 | 1.08 | - | 2 | 27 | 4 |
| D | Berlin—ERDF | −0.39 | 1.05 | - | 11 | 22 | 6 |
| SE | Upper Norrland—ERDF | −2.81 | 1.03 | - | 0 | 29 | 7 |
| FR | Île-de-France et Seine—ESF/ERDF/YEI | −19.60 | 1.03 | - | 1 | 41 | 8 |
| IT | Puglia—ERDF/ESF | 0.42 | 1.03 | - | 3 | 18 | 9 |
| IT | Umbria—ERDF | 0.88 | 1.02 | - | 26 | 13 | 10 |
| IT | Friuli-Venezia Giulia—ERDF | −17.78 | 1.02 | - | 1 | 40 | 11 |
| ES | Melilla—ERDF | −21.05 | 1.02 | - | 1 | 43 | 12 |
| FI | Sustainable growth and jobs—FI—ERDF/ESF | 0.50 | 1.01 | . | 4 | 15 | 13 |
| DE | Sachsen—ERDF | 0.98 | 1.01 | . | 7 | 11 | 14 |
| GR | Central Macedonia—ERDF/ESF | 0.9 | 1.01 | . | 13 | 12 | 15 |

Note: The grey cells identify the OPs that remain efficient regardless of the adjustments.

Figure 9 shows that the adjusted average funds dedicated to the eligible cost of efficient OPs were substantially reduced for efficient OPs (from EUR 75,514,840 to EUR 48,760,091), whereas they suffered a slight increase for inefficient ones (from EUR 16,621,816 to EUR 18,756,611). An analogous trend was obtained for the mean adjusted eligible spending and number of operations supported by efficient and inefficient OPs—see Figure 9.

Figure 10 suggests that if the effect of environmental factors is removed from the analysis, the number of operations supported is no longer the only factor that requires more attention from MAs. There are now 17 OPs that require a reduction in "eligible costs decided", i.e., a reduction in the amount of funding allotted to ICT in SMEs. In addition,

the enhancement of "eligible spending" is also an important barrier to efficiency for the same number of OPs.

The required adjustments that inputs and outputs must endure turning inefficient OPs into efficient ones were obtained with the adjusted factors (Table 14). These computations are also depicted in Figures 11 and 12 per OP, where OPs are portrayed in ascending order of efficiency from left to right. Overall, our findings suggest that, although the average adjusted factors depart from bigger values than previously (see Table 14) if we contrast the previous projections with the ones attained with the adjusted values, the average projection for the "total eligible spending" requires an enhancement of 42%, whereas the average projection for the "number of operations" supported only increases 4% and the average projection for the "total eligible cost decided" requires a further improvement of 9%.

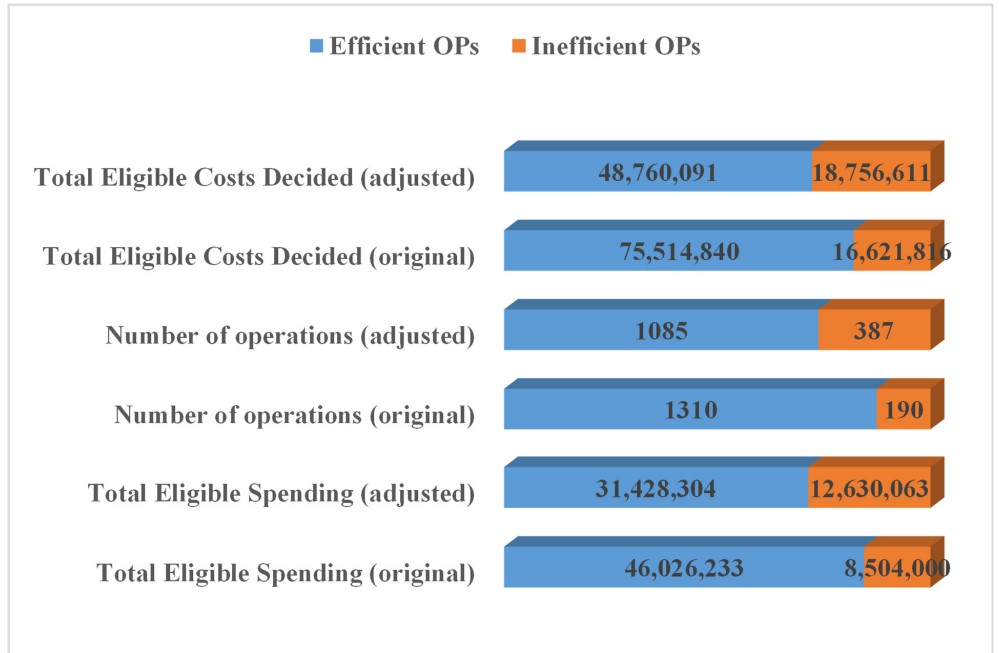

**Figure 9.** Average input and output values attained for the efficient and inefficient OPs with and without adjusted factors.

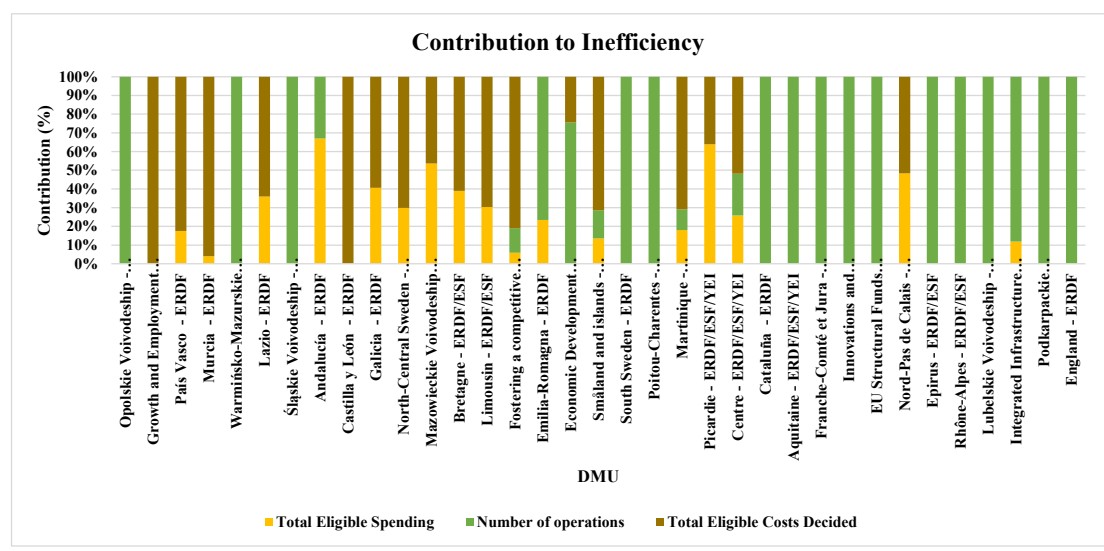

**Figure 10.** Contribution of inputs and outputs to inefficiency.

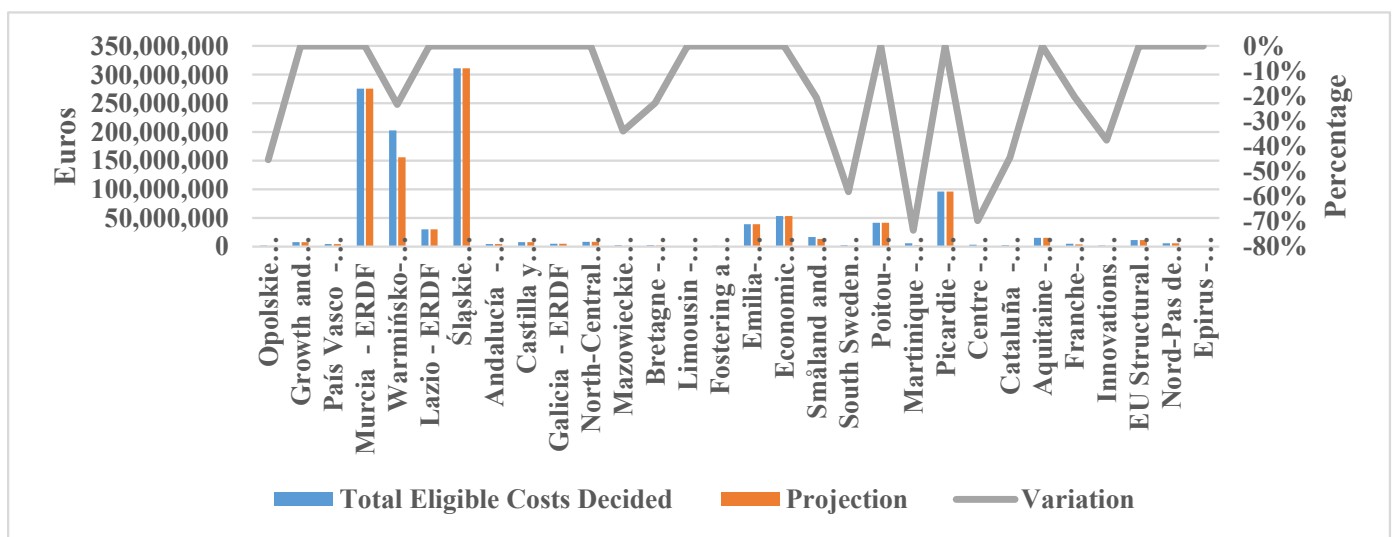

**Figure 11.** Potential output improvements for every inefficient OP.

**Figure 12.** Potential input improvement for every inefficient OP.

**Table 14.** Improvement potential for the OPs with and without adjusted factors.

| Factor | Average Adjusted | Average Original | Average Projection (Adjusted) | Average Projection (Original) |
|---|---|---|---|---|
| Total Eligible Spending | 12,630,063 | 8,504,000 | 14,050,185 | 9,912,457 |
| Number of Operations | 387.06 | 189.54 | 934.29 | 902.34 |
| Total Eligible Costs Decided | 18,756,611 | 16,621,816 | 16,602,992.51 | 15,284,716.33 |

Then again, the average "number of operations supported" shows the largest improvement potential of about 141% (i.e., it should increase on average from 387.06 to 934.29), while "eligible costs decided" (−11%) and "eligible spending" (11%) only require mild adjustments—see Table 14.

From the robustness analysis, it can be concluded that there are only three efficient OPs that remain efficient for data perturbations with both tolerances of 5% and 10%, which are "Provence-Alpes-Côte d'Azur—ERDF/ESF/YEI", "Enterprise and Innovation for Competitiveness—CZ—ERDF", and "Multi-regional Spain—ERDF". These OPs coincide with three of the previously most robust OPs without considering the adjusted factors. In addition, except for "England—ERDF" and "Integrated Infrastructure—SK—ERDF/CF" that become robustly inefficient for a tolerance of 5%, all the remaining OPs are potentially efficient for both tolerances.

## 6. Conclusions and Policy Implications

Since there are no studies available that specifically address the evaluation of the funds allocated to the OPs committed to ICT adoption by SMEs, this study is aimed at evaluating the procedural efficiency of the implementation of 51 OPs devoted to boosting ICT use in SMEs from 16 EU countries. Furthermore, because most of the methodologies used to evaluate ESIF in the context of ICT refer to ex post or ex ante assessments, we suggest a novel three-stage WRDD model framework that consists of combining this DEA model with SFA, which allows performing midterm/terminal assessments.

Our major findings concerning our research questions are given below.

RQ1. "Which indicators prevent the efficient utilization of ERDF allocated to boost ICT adoption in EU SMEs?"

Without the adjustment of the outputs, the "number of operations supported" is the indicator that requires more attention from MAs, whereas "eligible costs decided" and "eligible spending" only require attention in fewer cases (one and nine OPs, respectively). With the removal of the environmental factors and statistical noise, the "number of operations supported" is no more the single factor that demands more concern from MAs. With the adjusted factors, one-third of the OPs need to further reduce the "eligible costs decided". Furthermore, the improvement of "eligible spending" also becomes a relevant obstacle to efficiency for an identical number of OPs.

RQ2: "Which OPs were most frequently referenced as a source of best practices for the programmatic period under analysis?"

The OP that is more often selected as a benchmark regardless of the environmental factors is "Multi-regional Spain—ERDF", which is ranked among the four efficient OPs viewed as benchmarks in both scenarios. In addition, two (out of five) of the OPs that remain efficient regardless of the adjustments belong to Spanish regions. In this regard, it is worth mentioning that the use of "vouchers" in Spain [38] seems to be an efficient approach for reaching SMEs and providing them with help that is simple to administer and targeted toward their requirements.

RQ3: "Which OPs show more efficiency resilience in the face of probable changes in the indicators used?"

The robustness analysis shows that the OPs that remain efficient for data perturbations with both tolerances of 5% and 10% regardless of the adjustments are "Provence-Alpes-Côte d'Azur—ERDF/ESF/YEI", "Enterprise and Innovation for Competitiveness—CZ—ERDF", and "Multi-regional Spain—ERDF". Furthermore, "England—ERDF" and "Integrated Infrastructure—SK—ERDF/CF" become robustly inefficient for a tolerance of 5% either with or without the adjusted factors. Either way, these findings suggest that the management procedures of these OPs are the main reason behind these achievements.

RQ4: "Which environmental factors have the greatest impact on the inefficiency of the execution of OPs aimed at boosting ICT in EU SMEs?"

Results show that more developed regions and a higher rate of ICT specialists lead to an underuse of ERDF funds dedicated to boosting ICT in SMEs. In addition, a higher rate of ICT skills/specialists leads to the underachievement of the number of operations supported. On the other hand, more developed regions and a higher percentage of SMEs introducing product innovations seem to be more efficient in applying for funding.

These findings might be related to the acknowledged red tape procedures associated with the EU mechanisms, financing outlets, and bureaucratic regulations, namely for SMEs.

RQ5: "How does efficiency change with the removal of environmental factors?

With the adjusted factors, about 30% of the OPs (16) reached an efficient procedural performance against the previous 20% (10), highlighting the impacts of environmental factors on efficiency assessment. The regions that showed the largest increase in efficiency were "Melilla—ERDF", "Île-de-France et Seine—ESF/ERDF/YEI", and "Friuli-Venezia Giulia—ERDF", with values rising from −21.05 to 1.02 (105%), −19.60 to 1.03 (105%), and −17.78 to 1.02 (84%), respectively. These outcomes indicate that the inefficiencies of these OPs were mainly influenced by their external environment.

All in all, it can be ascertained that SMEs' access to ESIF (particularly ERDF) continues to be out of reach since they lack the administrative ability to deal with the various procedures for applying for and implementing ERDF projects. When opposed to "conventional" SME initiatives, this issue becomes more significant when it comes to ICT. In effect, increased speed and adaptability are required for initiatives in a sector known for rapid changes, such as ICT. Therefore, MAs should find ways of providing further support to SMEs that simplify administrative processes and address SMEs' specific needs.

Additionally, our study highlights one of the major limitations of this study, which is the lack of indicators available to assess the performance of the implementation of ESIF funds committed to ICT support in SMEs (only two outputs and one input were used for this purpose, despite the use of other environmental factors). As a result, national and regional policymakers should use supplemental specific indicators that account for ICT performance, tag spending that falls under other thematic objectives (rather than TO2) but has an ICT element, improve the quality and comprehensiveness of data on ICT performance at the regional and SME level, and consolidate distinct existing data sources.

Finally, while this study offered new perspectives and novel approaches to the efficiency evaluation of funding committed to boosting ICT use in European SMEs, future work should specifically assess the potential impacts of these ICT policy initiatives on economic indicators, such as GDP or employment, as well as their effect on territorial disparities.

**Author Contributions:** Writing—original draft, C.H. and C.V.; Revision and Final revision, C.H.; Methodology, C.V. and C.H.; Data curation, C.H. and C.V.; Conceptualization, C.H.; Validation, C.H.; Formal analysis, C.H.; Investigation, C.H.; Resources, C.H.; Project administration, C.H.; Statistical Analysis, C.V.; Validation and Final Revision, C.V. All authors have read and agreed to the published version of the manuscript.

**Funding:** This work has been funded by European Regional Development Fund in the framework of Portugal 2020—Programa Operacional Assistência Técnica (POAT 2020), under project POAT-01-6177-FEDER-000044 ADEPT: Avaliação de Políticas de Intervenção Co-financiadas em Empresas.

**Institutional Review Board Statement:** Not applicable.

**Informed Consent Statement:** Not applicable.

**Data Availability Statement:** Not applicable.

**Acknowledgments:** INESC Coimbra and CeBER are supported by the Portuguese Foundation for Science and Technology funds through Projects UID/MULTI/00308/2020 and UIDB/05037/2020, respectively.

**Conflicts of Interest:** The authors declare that they have no known competing financial interests or personal relationships that could have appeared to influence the work reported in this paper.

## Appendix A

**Table A1.** List of Cohesion Policy OPs' evaluations mentioning ICT and SMEs from 2015 to date.

| Title in English | Date of Publication | Authors | DR | LR | MD/DA | I | FG/FW | S | CS | B | E | SM | O |
|---|---|---|---|---|---|---|---|---|---|---|---|---|---|
| Ex-ante evaluation of financial instruments in Bourgogne-Franche-Comté, 2021–2027 | January 2021 | Francie Sadeski, Mathieu Boulestreau (francie.sadeski@technopolis-group.com) (accessed on 15 July 2022) | x | | | x | x | | | | | | |
| Evaluation of the implementation of the Bourgogne and Franche-Comté OPs, 2014–2020 | July 2017 | Téritéo Aster Europe Dictys conseil (contact@teriteo.fr) (accessed on 15 July 2022) | x | | x | x | | | | | | | x |
| Contribution to the Europe 2020 objectives of the ERDF and ESF Abruzzo OPs, 2014–2020 | February 2020 | Irs—Nomisma (www.irsonline.it-–-www.nomisma.it) (accessed on 15 July 2022) | x | | x | | | | | | | | |
| Annual evaluation of the Abruzzo ERDF OP 2014–2020 (2019) | July 2019 | IRS—Nomisma (IRS, www.irsonline.it-–-Nomisma, www.nomisma.it) (accessed on 15 July 2022) | | | x | x | | | | | | | |
| Evaluation of progress in implementing the EU Structural Funds Investments OP, 2014–2020 | December 2016 | PPMI Group (PPMI Group, GedimiNo av. 50, LT—01110 Vilnius, Lithuania, Tel.: +370-5-2620338, info@ppmi.lt, www.vpvi.lt/en) (accessed on 15 July 2022) | | | x | | | | | | | | x |
| Evaluation of financial instruments in Lithuania in 2014–2020 in both ERDF and ESF programmes | November 2017 | UAB "VG Consult" (accessed on 15 July 2022) | x | | | | | | x | | | x | |
| The impact of ESI Funds in 2014–2020 and the priorities for 2021–2027 | June 2017 | ESTEP Vilnius (Adress: Olimpiečių g. 1A-26, LT-09200 Vilnius; Phone number: 85269-0120; Fax: 85269-0124) (accessed on 15 July 2022) | | x | x | x | | | | | x | x | |
| Mid-term Evaluation of the Structural Funds Investment OP, 2014–2020 | March 2019 | ESTEP (Olimpiečių g. 1A-26, LT-09200 Vilnius, Tel.: +852690120, info@estep.lt, www.estep.lt) (accessed on 15 July 2022) | x | | x | x | | | | x | x | | |
| Evaluation of financing of Lithuanian economic sectors: post 2020 | June 2019 | ESTEP & PricewaterhouseCoopers (ESTEP Vilnius, Tel: +85269-0120, info@estep.lt, www.estep.lt -UAB PricewaterhouseCoopers, Tel.: +370-(5)-239-2300, vilnius@lt.pwc.com, www.pwc.com) (accessed on 15 July 2022) | | x | x | x | x | | x | | | | x |
| Evaluation of measures to reduce the digital divide under the Toscana ERDF OP, 2014–2020–2019 | July 2019 | Ecoter—Resco (http://www.ecoter.it/attivita.html-–-http://www.resco-ricerche.it/) (accessed on 15 July 2022) | | | x | x | | | | | | | x |
| Analysis of the market situation and suitable forms of support for the preparation of PA5 of the Technology and Applications for Competitiveness OP 2021–2027 | September 2020 | Association for European Funds (Association for European Funds, Budějovická 2056/96, 140 00 Prague 4, http://apef.cz/) (accessed on 15 July 2022) | | | x | | | | x | x | | | x |
| Analysis of the ICT Thematic Objective in 2014–2020 | December 2016 | Dirección General de Fondos Comunitarios (Directorate-General for Community Funds Programming and Evaluation GS) | | | x | | | | | | | | x |
| Follow-up monitoring evaluation of ICT support under the ERDF OPs, 2014–2020 | January 2020 | Ministerio de Hacienda (www.hacienda.gob.es) (accessed on 15 July 2022) | | | x | | | | | | | | x |
| Evaluation of the Cantabria ERDF OP, 2017 | May 2017 | Cantabria Government (GobierNo de Cantabria, C/Peña Herbosa 29, 39003 Santander, Cantabria, Tel.: +942-207-100) (accessed on 15 July 2022) | x | | x | x | | | x | | | | x |
| Evaluation of the Alsace, Lorraine and Champagne-Ardenne ESF/ERDF OPs, 2014–2020 | June 2018 | Téritéo and Aster (sabourin@teriteo.fr) (accessed on 15 July 2022) | | | x | | | | | | | | |

**Table A1.** *Cont.*

| Title in English | Date of Publication | Authors | Methods [a] | | | | | | | | | | |
|---|---|---|---|---|---|---|---|---|---|---|---|---|---|
| | | | DR | LR | MD/DA | I | FG/FW | S | CS | B | E | SM | O |
| Evaluation of High Technology Network in Emilia Romagna, 2014–2020 | December 2019 | IRS- Nomisma (www.irsonline.it---Via Castiglione 4, 40124 Bologna—irsbo@irsonline.it; nomisma.it—Strada Maggiore, 44—40125 Bologna—info@nomisma.it) (accessed on 15 July 2022) | x | | x | | | | x | | | x | |
| Evaluation of the Regional OP 2014–2020 technology transfer interventions | August 2019 | Fabrizio Tenna—Lattanzio Advisory Spa (www.lattanziokibs.com/ro) (accessed on 15 July 2022) | | x | | x | x | | x | x | | x | |
| Progress report on the Abruzzo ESF OP, 2014–2020 | December 2018 | IRS—NOMISMA (www.irsonline.it) | x | | x | | | | | | | | |
| Evaluation of Specific Objectives 3.5, 3.6 and 4.2 of the Enterprise and Innovation for Competitiveness OP, 2014–2020, in the Czech Republic | July 2019 | Daniel Mayer (Association for European Funds; www.apef.cz) (accessed on 15 July 2022) | x | | x | | | | x | | | | |
| Annual evaluation report (2019) of the Toscana ESF OP, 2014–2020 | June 2020 | Ismeri Europa (www.ismerieuropa.com) (accessed on 15 July 2022) | x | | x | | | | | | | | |
| Evaluation report on implementation procedures and their digitalisation in the Bolzano ERDF OP, 2014–2020 | January 2020 | IRS—PTS clas (www.irsonline.it) (accessed on 15 July 2022) | x | | x | x | | | x | | | | |
| Implementation of TO 11 and 2 in Italian OPs in 2014–2020 | May 2017 | Segreteria Tecnica comitato pilotaggio (accessed on 15 July 2022) | x | | x | | | | x | x | | | |

Note: [a] DR—desk research; LR—literature review; MD/DA—monitoring data/data analysis; I—interviews; FG/FW—focus groups/facilitated workshops; S—surveys; CS—case studies; B—benchmarking; E—expert consultation; SM—statistical methods; O—other.

**Table A2.** Data on the inputs and outputs.

| MS (2 Digit ISO) | Program Title | DMU | Output | Output | Input |
|---|---|---|---|---|---|
| | | | Total Eligible Spending | Number of Operations | Total Eligible Costs Decided |
| ES | Andalucía—ERDF | 1 | 6,695,517 | 649 | 21,370,083 |
| FR | Aquitaine—ERDF/ESF/YEI | 2 | 13,534,588 | 99 | 17,402,571 |
| DE | Berlin—ERDF | 3 | 876,173 | 10 | 876,173 |
| FR | Bretagne—ERDF/ESF | 4 | 1,175,044 | 6 | 1,502,834 |
| ES | Castilla y León—ERDF | 5 | 3,356,883 | 66 | 3,478,014 |
| ES | Cataluña—ERDF | 6 | 3,233,273 | 3 | 7,401,115 |
| GR | Central Macedonia—ERDF/ESF | 7 | 3,860,885 | 410 | 4,538,729 |
| FR | Centre—ERDF/ESF/YEI | 8 | 1,763,215 | 14 | 3,540,598 |
| GR | Competitiveness Entrepreneurship and Innovation—GR—ERDF/ESF | 9 | 100,667,978 | 5457 | 275,856,182 |
| HU | Economic Development and Innovation Programme—HU—ERDF/ESF/YEI | 10 | 102,175,668 | 1271 | 202,847,237 |
| IT | Emilia-Romagna—ERDF | 11 | 9,781,930 | 725 | 30,248,825 |
| UK | England—ERDF | 12 | 41,078,593 | 18 | 74,764,788 |
| CZ | Enterprise and Innovation for Competitiveness—CZ—ERDF | 13 | 237,904,467 | 451 | 311,154,920 |
| GR | Epirus—ERDF/ESF | 14 | 4,593,855 | 144 | 4,593,855 |
| LT | EU Structural Funds Investments—LT—ERDF/ESF/CF/YEI | 15 | 7,607,793 | 210 | 7,786,656 |
| ES | Extremadura—ERDF | 16 | 1,560,112 | 810 | 4,823,735 |
| MT | Fostering a competitive and sustainable economy—MT—ERDF/CF | 17 | 182,337 | 1 | 5,000,000 |

**Table A2.** *Cont.*

| MS (2 Digit ISO) | Program Title | DMU | Output | Output | Input |
|---|---|---|---|---|---|
| | | | **Total Eligible Spending** | **Number of Operations** | **Total Eligible Costs Decided** |
| FR | Franche-Comté et Jura—ERDF/ESF | 18 | 5,982,231 | 6 | 8,210,556 |
| IT | Friuli-Venezia Giulia—ERDF | 19 | 248,601 | 1 | 671,429 |
| ES | Galicia—ERDF | 20 | 760,336 | 2 | 2,112,350 |
| LV | Growth and Employment—LV—ERDF/ESF/CF/YEI | 21 | 1,394,604 | 6 | 2,318,306 |
| FR | Haute-Normandie—ERDF/ESF/YEI | 22 | 243,167 | 2 | 373,794 |
| FR | Île-de-France et Seine—ESF/ERDF/YEI | 23 | 847,778 | 2 | 1,296,717 |
| BG | Innovations and Competitiveness—BG—ERDF | 24 | 33,942,154 | 228 | 38,612,352 |
| SK | Integrated Infrastructure—SK—ERDF/CF | 25 | 13,281,761 | 211 | 52,773,554 |
| IT | Lazio—ERDF | 26 | 1,906,215 | 733 | 16,669,118 |
| FR | Limousin—ERDF/ESF | 27 | 966,418 | 17 | 2,679,334 |
| PL | Lubelskie Voivodeship—ERDF/ESF | 28 | 21,893,553 | 85 | 41,255,981 |
| FR | Martinique—ERDF/ESF/YEI | 29 | 814,608 | 14 | 5,637,368 |
| PL | Mazowieckie Voivodeship—ERDF/ESF | 30 | 813,198 | 3 | 1,712,365 |
| ES | Melilla—ERDF | 31 | 68,486 | 2 | 1,170,261 |
| ES | Multi-regional Spain—ERDF | 32 | 58,864,158 | 5108 | 95,971,219 |
| ES | Murcia—ERDF | 33 | 1,963,414 | 155 | 3,167,696 |
| FR | Nord-Pas de Calais—ERDF/ESF/YEI | 34 | 340,394 | 2 | 3,281,904 |
| SE | North-Central Sweden—ERDF | 35 | 1,959,890 | 3 | 2,092,127 |
| PL | Opolskie Voivodeship—ERDF/ESF | 36 | 9,800,608 | 152 | 15,037,690 |
| ES | País Vasco—ERDF | 37 | 3,964,897 | 575 | 4,618,616 |
| FR | Picardie—ERDF/ESF/YEI | 38 | 311,847 | 6 | 1,632,728 |
| PL | Podkarpackie Voivodeship—ERDF/ESF | 39 | 10,827,667 | 120 | 11,479,570 |
| FR | Poitou-Charentes—ERDF/ESF | 40 | 3,238,795 | 22 | 5,495,141 |
| FR | Provence-Alpes-Côte d'Azur—ERDF/ESF/YEI | 41 | 329,249 | 1 | 251,294 |
| IT | Puglia—ERDF/ESF | 42 | 336,829 | 27 | 701,493 |
| FR | Rhône-Alpes—ERDF/ESF | 43 | 3,937,528 | 6 | 9,346,524 |
| DE | Sachsen—ERDF | 44 | 34,693,199 | 2184 | 51,464,402 |
| PL | Śląskie Voivodeship—ERDF/ESF | 45 | 20,359,931 | 218 | 27,139,088 |
| SE | Småland and islands—ERDF | 46 | 3,081,019 | 12 | 4,901,930 |
| SE | South Sweden—ERDF | 47 | 3,606,572 | 9 | 8,014,161 |
| FI | Sustainable growth and jobs—FI—ERDF/ESF | 48 | 21,515,886 | 394 | 29,109,235 |
| IT | Umbria—ERDF | 49 | 1,229,575 | 120 | 1,492,122 |
| SE | Upper Norrland—ERDF | 50 | 135,762 | 2 | 409,753 |
| PL | Warmińsko-Mazurskie Voivodeship—ERDF/ESF | 51 | 5,217,675 | 103 | 8,356,361 |

**Table A3.** Data on the environmental factors.

| Member State | Original | S1 (Eligible Spending) | S2 (Number of Operations) | GDP PPP PC | Population with Tertiary Education | Digital Skills | R&D Expenditures Business Sector | ICT Specialists | Product Process Innovators |
|---|---|---|---|---|---|---|---|---|---|
| ES | Andalucía—ERDF | 7,336,341.74 | 757.22 | 66.80 | 0.53 | 0.62 | 0.12 | 0.25 | 0.20 |
| FR | Aquitaine—ERDF/ESF/YEI | 0.00 | 708.41 | 88.41 | 0.62 | 0.52 | 0.29 | 0.23 | 0.62 |
| DE | Berlin—ERDF | 0.00 | 55.52 | 123.58 | 0.80 | 0.72 | 0.51 | 1.00 | 0.94 |
| FR | Bretagne—ERDF/ESF | 0.00 | 175.33 | 89.18 | 0.75 | 0.53 | 0.44 | 0.39 | 0.67 |
| ES | Castilla y León—ERDF | 0.00 | 134.21 | 84.83 | 0.79 | 0.62 | 0.31 | 0.23 | 0.27 |
| ES | Cataluña—ERDF | 0.00 | 119.80 | 106.87 | 0.86 | 0.67 | 0.34 | 0.46 | 0.36 |
| GR | Central Macedonia—ERDF/ESF | 37,508.77 | 154.50 | 52.20 | 0.64 | 0.35 | 0.15 | 0.19 | 1.00 |
| FR | Centre—ERDF/ESF/YEI | 0.00 | 524.67 | 86.81 | 0.55 | 0.50 | 0.42 | 0.18 | 0.65 |
| HU | Economic Development and Innovation Programme—HU—ERDF/ESF/YEI | 0.00 | 2710.43 | 70.78 | 0.20 | 0.36 | 0.44 | 0.43 | 0.33 |
| IT | Emilia-Romagna—ERDF | 7,762,830.87 | 1283.90 | 117.99 | 0.44 | 0.35 | 0.56 | 0.40 | 0.86 |
| UK | England—ERDF | 4,453,169.59 | 4090.02 | 102.49 | 0.69 | 0.93 | 0.46 | 0.65 | 0.53 |
| MT | Fostering a competitive and sustainable economy—MT—ERDF/CF | 1,488,592.08 | 817.31 | 98.76 | 0.50 | 0.66 | 0.13 | 0.66 | 0.64 |
| FR | Franche-Comté et Jura—ERDF/ESF | 0.00 | 759.80 | 83.83 | 0.57 | 0.49 | 0.47 | 0.18 | 0.59 |
| IT | Friuli-Venezia Giulia—ERDF | 193,744.84 | 74.33 | 104.10 | 0.40 | 0.34 | 0.32 | 0.41 | 0.67 |
| ES | Galicia—ERDF | 69,893.76 | 328.28 | 81.14 | 0.79 | 0.61 | 0.17 | 0.36 | 0.24 |
| LV | Growth and Employment—LV—ERDF/ESF/CF/YEI | 0.00 | 319.61 | 67.38 | 0.60 | 0.34 | 0.05 | 0.38 | 0.25 |
| FR | Haute-Normandie—ERDF/ESF/YEI | 119,057.98 | 20.67 | 86.40 | 0.47 | 0.48 | 0.37 | 0.16 | 0.54 |
| FR | Île-de-France et Seine—ESF/ERDF/YEI | 0.00 | 164.81 | 178.30 | 1.00 | 0.58 | 0.71 | 1.00 | 0.79 |
| SK | Integrated Infrastructure—SK—ERDF/CF | 18,424,203.78 | 2860.04 | 72.37 | 0.46 | 0.41 | 0.17 | 0.49 | 0.18 |
| IT | Lazio—ERDF | 11,366,082.33 | 0.00 | 112.36 | 0.44 | 0.34 | 0.23 | 0.99 | 0.79 |
| FR | Limousin—ERDF/ESF | 16,439.13 | 413.59 | 87.74 | 0.78 | 0.51 | 0.80 | 0.48 | 0.65 |
| PL | Lubelskie Voivodeship—ERDF/ESF | 2,571,361.29 | 2442.94 | 49.09 | 0.64 | 0.29 | 0.13 | 0.16 | 0.24 |
| FR | Martinique—ERDF/ESF/YEI | 1,257,031.70 | 834.37 | 73.52 | 0.37 | 0.43 | 0.02 | 0.21 | 0.51 |
| PL | Mazowieckie Voivodeship—ERDF/ESF | 0.00 | 249.19 | 60.81 | 0.57 | 0.31 | 0.13 | 0.05 | 0.25 |
| ES | Melilla—ERDF | 508,141.26 | 161.59 | 66.37 | 0.42 | 0.64 | 0.02 | 0.23 | 0.20 |
| ES | Murcia—ERDF | 0.00 | 293.41 | 74.59 | 0.52 | 0.64 | 0.17 | 0.13 | 0.24 |
| FR | Nord-Pas de Calais—ERDF/ESF/YEI | 804,669.95 | 535.20 | 83.54 | 0.66 | 0.52 | 0.24 | 0.32 | 0.64 |
| SE | North-Central Sweden—ERDF | 0.00 | 175.27 | 97.31 | 0.54 | 0.84 | 0.33 | 0.40 | 0.68 |
| PL | Opolskie Voivodeship—ERDF/ESF | 0.00 | 977.87 | 60.17 | 0.53 | 0.31 | 0.17 | 0.47 | 0.22 |
| FR | Picardie—ERDF/ESF/YEI | 389,272.52 | 239.42 | 83.54 | 0.66 | 0.52 | 0.24 | 0.32 | 0.64 |
| FR | Poitou-Charentes—ERDF/ESF | 0.00 | 707.91 | 88.41 | 0.62 | 0.52 | 0.29 | 0.23 | 0.62 |
| IT | Puglia—ERDF/ESF | 113,609.82 | 53.65 | 62.46 | 0.13 | 0.28 | 0.11 | 0.18 | 0.69 |
| FR | Rhône-Alpes—ERDF/ESF | 466,042.73 | 118.44 | 101.97 | 0.85 | 0.53 | 0.65 | 0.42 | 0.75 |
| DE | Sachsen—ERDF | 0.00 | 166.87 | 86.19 | 0.12 | 0.71 | 0.16 | 0.20 | 0.61 |
| PL | Śląskie Voivodeship—ERDF/ESF | 0.00 | 865.07 | 74.67 | 0.64 | 0.30 | 0.18 | 0.33 | 0.25 |
| SE | Småland and islands—ERDF | 0.00 | 670.80 | 104.24 | 0.59 | 0.90 | 0.50 | 0.43 | 0.89 |
| SE | South Sweden—ERDF | 0.00 | 947.83 | 105.49 | 0.82 | 0.85 | 0.72 | 0.80 | 0.98 |
| FI | Sustainable growth and jobs—FI—ERDF/ESF | 0.00 | 792.80 | 111.10 | 0.55 | 0.93 | 0.74 | 1.00 | 0.77 |
| IT | Umbria—ERDF | 132,621.38 | 44.08 | 85.03 | 0.38 | 0.33 | 0.18 | 0.41 | 0.69 |
| SE | Upper Norrland—ERDF | 236,142.84 | 27.04 | 114.95 | 0.61 | 0.88 | 0.18 | 0.49 | 0.80 |
| PL | Warmińsko-Mazurskie Voivodeship—ERDF/ESF | 0.00 | 745.82 | 50.55 | 0.36 | 0.31 | 0.09 | 0.16 | 0.18 |

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
