# Peer review of "How Efficient Is the Implementation of Structural Funds Committed to Enhancing ICT Adoption in SMEs?"

_2199-8531, doi:10.3390/joitmc8030147_

Round 1
Reviewer 1 Report
I think that the paper deals with an interesting and hotely topic. It explains well the used methodology, that it is appropriate. The paper includes a good literature review, being the main contribution the empirical analysis.
The literature is appropriate and up to date.
The results are relevant and the conclusions are well founded in the research. I also appreciate the policy implications of the approach.
I only suggest a minor change, explain from the introduction the spatial area analysed in this research.
Author Response
Reviewer 1
I think that the paper deals with an interesting and hotely topic. It explains well the used methodology, that it is appropriate. The paper includes a good literature review, being the main contribution the empirical analysis.
The literature is appropriate and up to date.
The results are relevant and the conclusions are well founded in the research. I also appreciate the policy implications of the approach.
R: We would like to thank the reviewer for his/her positive comments.
I only suggest a minor change, explain from the introduction the spatial area analysed in this research.
R: Thank you very much for your valuable suggestion. With this regard we have introduced the following information in the Introduction (note that all the abbreviations were defined previously in the text):
“We propose a novel three-stage WRDD model in combination with SFA to evaluate 51 OPs from 16 EU MS.”

Reviewer 2 Report
The paper analyses important topic- efficiency of implementation of structural funds committed to enhancing ICT adoption in SMEs. The paper applies relevant advanced research methods like a novel three-stage Weighted Russel Directional Distance (WRDD) data envelopment analysis (DEA) model in conjunction with Stochastic Frontier Analysis (SFA). It includes the wide literature review and comprehensive discussion and analysis of research results in line with other studies. I think paper can be published after minor revision. The authors need to stress the input of this paper. They should make conclusions more concise. The limits of this study should be clearly identified and future research guidelines need to be provided.
Author Response
Reviewer 2
The paper analyses important topic-efficiency of implementation of structural funds committed to enhancing ICT adoption in SMEs. The paper applies relevant advanced research methods like a novel three-stage Weighted Russel Directional Distance (WRDD) data envelopment analysis (DEA) model in conjunction with Stochastic Frontier Analysis (SFA). It includes the wide literature review and comprehensive discussion and analysis of research results in line with other studies. I think paper can be published after minor revision.
R: We would like to thank the reviewer for his/her positive comments.
The authors need to stress the input of this paper.
R: Thank you very much for your valuable suggestion. With this regard we have introduced the following information in the Conclusions:
“Since there are no studies available that specifically address the evaluation of the funds allocated to the OPs committed to ICT adoption by SMEs, this study is aimed at evaluating the procedural efficiency of the implementation of 51 OPs devoted to boosting ICT use in SMEs from 16 EU countries. Furthermore, because most of the methodologies used to evaluate ESIF in the context of ICT refer to ex-post or ex-ante assessments, we suggest a novel three-stage WRDD model framework that consists of combining this DEA model with SFA, which allows performing midterm/terminal assessments.”
They should make conclusions more concise.
R: Thank you very much for your valuable suggestion. With this regard we significantly shortened the conclusions and the Section of the methodology.
The limits of this study should be clearly identified and future research guidelines need to be provided.
R: Thank you very much for your valuable suggestion. With this regard we introduced the following information in the Conclusions:
“Additionally, our study highlights one of the major limitations of this study, which is the lack of indicators available to assess the performance of the implementation of ESIF funds committed to ICT support in SMEs (only two outputs and one input were used for this purpose, despite the use of other environmental factors). As a result, national and regional policymakers should use supplemental specific indicators that account for ICT performance; tag spending that falls under other thematic objectives (rather than TO2) but has an ICT element; improve the quality and comprehensiveness of data on ICT performance at the regional and SME level; and consolidate distinct existing data sources.
Finally, while this study offered new perspectives and novel approaches to the efficiency evaluation of funding committed to boosting ICT use in European SMEs, future work should specifically assess the potential impacts of these ICT policy initiatives on economic indicators, such as GDP or employment, as well as their effect on territorial disparities.”

Reviewer 3 Report
The manuscript aims to evaluated the execution of the operational programmes (OPs) committed to encouraging the adoption of information and communication technologies (ICT) in small medium-sized enterprises (SMEs).
The article is too long and dispersed
The manuscript needs more review of literature
The literature review (paragraph 2) can be extend with other recent works, because the question of whether and how innovation can be replicated and applied in a wider context is strictly connected to the understanding of those factors and mechanisms capable of determining the success or failure of the introduction of innovation itself.
See and cite for example the following work.
Fanelli R. M. (2021). Barriers to Adopting New Technologies within Rural Small and Medium Enterprises (SMEs). Social Sciences, 10(11): 430. https://doi.org/10.3390/socsci10110430.
…and consider others references reported at the end of this manuscript.
3 (4) Materials and Methods
The description of the methodological approach is too long; I suggest summarizing it.
4 (3). I suggest moving this paragraph and putting it before paragraph 3
5. Analysis and discussion of results
The results and findings should be discussed and compared with previous studies and findings.
This paragraph is too long and dispersive; I suggest summarizing it.
6. Conclusions and policy implications
Conclusions and recommendations should be improved as they largely repeated the results. The character of conclusion is too general one and it repeats results. Authors should better underline conclusions, and intensions for future researches should be noted at the end of the conclusions.
I suggest that you report concise answers to the four RQs
RQ1. "Which indicators prevent the efficient utilisation of ERDF allocated to boost ICT adoption in EU SMEs?"
RQ2: "Which OPs were most frequently referenced as a source of best practices for the programmatic period under analysis?"
RQ3: "Which OPs show more efficiency resilience in the face of probable changes in the indicators used?"
RQ4: "Which environmental factors have the greatest impact on the inefficiency of the execution of OPs aimed at boosting ICT in EU SMEs?"
At the end, I suggest to improve di quality of tables and figures
Author Response
Reviewer 3
The manuscript aims to evaluated the execution of the operational programmes (OPs) committed to encouraging the adoption of information and communication technologies (ICT) in small medium-sized enterprises (SMEs).
The article is too long and dispersed
R: Thank you very much for your valuable Observation. In this regard, we have shortened the Conclusions and the Section of the Methodology.
The manuscript needs more review of literature
The literature review (paragraph 2) can be extend with other recent works, because the question of whether and how innovation can be replicated and applied in a wider context is strictly connected to the understanding of those factors and mechanisms capable of determining the success or failure of the introduction of innovation itself.
See and cite for example the following work.
Fanelli R. M. (2021). Barriers to Adopting New Technologies within Rural Small and Medium Enterprises (SMEs). Social Sciences, 10(11): 430. https://doi.org/10.3390/socsci10110430.
…and consider others references reported at the end of this manuscript.
R: Thank you very much for your valuable suggestion. With regard to this observation, we would like to stress that this paper has been revised be three reviewers.
The first reviewer states: “The literature is appropriate and up to date.”
The second reviewer states: “It includes the wide literature review and comprehensive discussion and analysis of research results in line with other studies”.
Therefore, despite considering this suggestion, i.e., we have added the aforementioned citation (in the literature mentioned in the Introduction and in the corroboration of some of the results obtained) and another paper of that same manuscript (see the paper reference introduced below), we would like to stress that this paper does not specifically address rural SMEs, nor it accounts for the assessment of the European Agricultural Fund for Rural Development (EAFRD). Therefore, we have only inserted these two additional references to avoid dispersion.
References added:
Fanelli, R.M. Barriers to Adopting New Technologies within Rural Small and Medium Enterprises (SMEs). Social Sciences 2021, 10, 430, doi:10.3390/socsci10110430.
- Zhang, M.; Hartley, J.L. Guanxi, IT Systems, and Innovation Capability: The Moderating Role of Proactiveness. Journal of Business Research 2018, 90, 75–86, doi:10.1016/j.jbusres.2018.04.036.
3 (4) Materials and Methods
The description of the methodological approach is too long; I suggest summarizing it.
R: Thank you very much for your valuable suggestion. Our section is not called Materials and methods, so we assume that the reviewer is referring to the Section “3. Methodology”. In this regard we shortened this section as suggested.
4 (3). I suggest moving this paragraph and putting it before paragraph 3
R: Thank you very much for your observation. We are assuming that you are referring to Section 4, paragraph 3. If this is the case, paragraph 3 is now paragraph 2. Please check the manuscript to confirm.
- Analysis and discussion of results
The results and findings should be discussed and compared with previous studies and findings.
R: Thank you very much for your valuable suggestion. In this regard, we would like to remind the reviewer that to the best of our knowledge this is the first and only study assessing the OPs devoted to foster the adoption of ICT in EU SMEs. Therefore, the literature on this topic is scarce as acknowledged by Reggi and Gil-Garcia (2021).
Nevertheless, we have contrasted, when possible, our results with the previous literature. Please see the parts of the text below.
“The four OPs that are most frequently selected as benchmarks are “Extremadura - ERDF” (30), “País Vasco - ERDF” (25), “Multi-regional Spain - ERDF” (22) and “Provence-Alpes-Côte d'Azur - ERDF/ESF/YEI” (21) – see Table 7. Curiously, all these regions are located in countries viewed as being the highest spenders on ICT support for SMEs (Pellegrin et al., 2018). Furthermore, the findings obtained by Spanish regional OPs (the country with the OPs most frequently selected as benchmarks) match the conclusions of Ruiz-Rodríguez et al. (2018). In line with these authors, Spanish regions at the firm level are in a moderate or greater level of digital advancement than their EU peers, as well as displaying a reduced DD among them (i.e., a smaller degree of difference in firm digital development) in com-parison to what occurs in the remaining European MS.”
(…)
“All in all, these outcomes suggest a poor application of ERDF funds in ICT SME support. These findings appear to be consistent with the challenges identified by Pellegrin et al. (2018) at the heart of the EU ICT policy in terms of digital achievements. According to their findings, the EU seems to underachieve its competitors (specifically, the United States, Japan, and South Korea) in terms of innovative ICT infrastructures and ICT adoption, in particular by SMEs. In fact, Fanelli (2021) ascertained that to create the encouraging circumstances and provide incentives to SMEs for adopting innovative solutions, SMEs need explicit policy initiatives to guarantee that technological services can be supplied and that the required infrastructures are accessible.”
(…)
“These findings suggest that more developed regions and a higher rate of ICT specialists do not necessarily mean a higher absorption rate of ERDF funds dedicated to boosting ICT in SMEs. The underuse of ERDF by ICT Croatian SMEs was also acknowledged by Bukvić et al. (2021) for the period of 2014-2020. These authors concluded that the intricacy and the required time to apply, develop and appraise the projects were the potential cause of these findings (Bukvić et al., 2021). Also, Martinez-Cillero et al. (2020) ascertained that SMEs’ investment ends up being smaller than it would be anticipated by standard economic fundaments, indicating that these firms are particularly vulnerable to funding issues. Another reason for these findings might reflect the use of other sources of financing by these SMEs (Pellegrin et al., 2018).”
(…)
“Also, 2 (out of 5) of the OPs that remain efficient regardless of the adjustments belong to Spanish regions (see grey cells of Table 12). In this context, it is worth mentioning that MS in Southern (such as Italy and Spain) and Central and Eastern European countries were the biggest recipients of financing for ICT and the digital economy (Pellegrin et al., 2018). This is particularly true for countries with efficient OPs, such as Spain, Greece, Hungary, the Czech Republic, and the Baltic States (Pellegrin et al., 2018).”
This paragraph is too long and dispersive; I suggest summarizing it.
R: Thank you very much for your suggestion. However, with the information provided by the reviewer it was not possible to identify the paragraph.
Conclusions and policy implications
Conclusions and recommendations should be improved as they largely repeated the results. The character of conclusion is too general one and it repeats results. Authors should better underline conclusions, and intensions for future researches should be noted at the end of the conclusions.
I suggest that you report concise answers to the four RQs
RQ1. "Which indicators prevent the efficient utilisation of ERDF allocated to boost ICT adoption in EU SMEs?"
RQ2: "Which OPs were most frequently referenced as a source of best practices for the programmatic period under analysis?"
RQ3: "Which OPs show more efficiency resilience in the face of probable changes in the indicators used?"
RQ4: "Which environmental factors have the greatest impact on the inefficiency of the execution of OPs aimed at boosting ICT in EU SMEs?"
R: Thank you very much for your suggestion. We have shortened the Conclusions accordingly and we have just answered to the 5 research questions.
Besides, we have specifically identified the main contributions of our work:
“Since there are no studies available that specifically address the evaluation of the funds allocated to the OPs committed to ICT adoption by SMEs, this study is aimed at evaluating the procedural efficiency of the implementation of 51 OPs devoted to boosting ICT use in SMEs from 16 EU countries. Furthermore, because most of the methodologies used to evaluate ESIF in the context of ICT refer to ex-post or ex-ante assessments, we suggest a novel three-stage WRDD model framework that consists of combining this DEA model with SFA, which allows performing midterm/terminal assessments.”
Finally, future work developments were also suggested:
“Finally, while this study offered new perspectives and novel approaches to the efficiency evaluation of funding committed to boosting ICT use in European SMEs, future work should specifically assess the potential impacts of these ICT policy initiatives on economic indicators, such as GDP or employment, as well as their effect on territorial disparities.”
At the end, I suggest to improve di quality of tables and figures
R: Thank you very much for your suggestion. We have improved the quality of the tables and figures.
